

# Spaceborne differential absorption radar water vapor retrieval capabilities in tropical and subtropical boundary layer cloud regimes

Richard J. Roy, Matthew Lebsock, and Marcin Kurowski

Jet Propulsion Laboratory, California Institute of Technology, Pasadena, California, USA

**Correspondence:** Matthew Lebsock (matthew.d.lebsock@jpl.nasa.gov)

**Abstract.** Differential absorption radar (DAR) near the 183 GHz water vapor absorption line is an emerging measurement technique for humidity profiling inside of clouds and precipitation with high vertical resolution, as well as for measuring integrated water vapor (IWV) in clear air regions. For radar transmit frequencies on the water line flank away from the highly attenuating line center, the DAR system becomes most sensitive to water vapor in the planetary boundary layer (PBL), which is a region of

the atmosphere that is poorly resolved in the vertical by existing spaceborne humidity and temperature profiling instruments. In this work, we present a high-fidelity, end-to-end simulation framework for notional spaceborne DAR instruments that feature realistically achievable radar performance metrics, and apply this simulator to assess DAR's PBL humidity observation capabilities. Both the assumed instrument parameters and radar retrieval algorithm leverage recent technology and algorithm development for an existing airborne DAR instrument. To showcase the capabilities of DAR for humidity observations in a

variety of relevant PBL settings, we implement the instrument simulator in the context of large eddy simulations (LES) of 5 different cloud regimes throughout the trade-wind subtropical-to-tropical cloud transition. Three distinct DAR humidity observations are investigated: IWV between the top of the atmosphere and the first detected cloud bin or Earth's surface; in-cloud water vapor profiles with 200 meter vertical resolution; and IWV between the last detected cloud bin and the Earth's surface, which can provide a precise measurement of the sub-cloud humidity. We provide a thorough assessment of the systematic and

random errors for all 3 measurement products for each LES case, and analyze the humidity precision scaling with along-track measurement integration. While retrieval performance depends greatly on the specific cloud regime, we find generally that for a radar with cross-track scanning capability, in-cloud profiles with 200 m vertical resolution and 10-20% uncertainty can be retrieved for horizontal integration distances of 100-200 km. Furthermore, column IWV can be retrieved with 10% uncertainty for 10-20 km of horizontal integration. Finally, we provide some example science applications of the simulated DAR obser-

vations, including estimating near-surface relative humidity using the cloud-to-surface column IWV, and inferring in-cloud temperature profiles from the DAR water vapor profiles by assuming a fully saturated environment.



# 1    Introduction

Cloud morphology and precipitation depend sensitively on the three-dimensional distributions of water vapor and temperature, especially within the planetary boundary layer (PBL) where most convective initiation occurs. Existing spaceborne sensors have limited ability to sample water vapor and temperature in the PBL with high spatial resolution, with increased

difficulty inside of cloudy and precipitating volumes for passive infrared and microwave sounders (Wulfmeyer et al., 2015; Stevens et al., 2017; Sahoo et al., 2015). The 2017 Earth Science Decadal Survey (National Academies of Sciences, Engineering, and Medicine (NASEM), 2018) has recommended the incubation of technologies that enable improved spaceborne measurements of PBL thermodynamics during the current decade. Active humidity sounding approaches, including differential absorption lidar (DIAL) and differential absorption radar (DAR), offer new potential spaceborne solutions for providing

high-vertical-resolution water vapor profiles in clear sky and cloudy regions, respectively (Nehrir et al., 2017), with a typical vertical resolution target of 200 m. However, because DAR is a relatively new measurement approach with limited instrument deployment (Roy et al., 2020, 2021; Cooper et al., 2020) and simulation (Lebsock et al., 2015; Millán et al., 2016; Battaglia and Kollias, 2019; Millán et al., 2020) heritage, there is a need to critically assess the measurement capabilities of notional spaceborne DAR systems with detailed orbital simulations, in addition to continued assessment of observational capabilities

from airborne platforms.

In this work, we expand significantly on previous spaceborne DAR instrument simulation efforts to provide a detailed assessment of DAR retrieval capabilities in the context of different regimes throughout the trade-wind marine cloud transition from stratocumulus to deep convection. First, we utilize 5 different large-eddy simulations (LES) that represent distinct cloud regimes throughout the transition, and perform forward radar simulations that include detailed effects such as realistic electro-

magnetic scattering from ice particles, non-uniform beam filling leveraging the high spatial resolution of the LES, and multiple scattering. Second, we build on retrieval approaches developed previously for real DAR observations (Roy et al., 2020) and implement a new least-squares retrieval algorithm that mitigates retrieval bias using an improved spectral fitting function and retrieves the entire vertical humidity profile as part of a single optimization procedure. The new formulation employs an improved humidity interpolation function that allows for a flexible retrieval of water vapor between arbitrarily spaced cloud range

bins, which is important given the inherently sparse sampling of radar reflectivity profiles. The result is a seamless retrieval of both high-vertical-resolution (200 m) in-cloud profiles *and* integrated water vapor measurements in clear air columns (e.g., between the cloud base and the surface). Importantly, this retrieval does not require specification of a prior distribution for the humidity profile, nor does it impose any regularizing constraints on the form of the retrieved profile as in Roy et al. (2020). The linearity of the retrieval algorithm's forward model in the state vector allows for a straightforward transformation of random

measurement error to retrieved state uncertainty, which is then analyzed as a function of along-track averaging distance to assess the achievable humidity precision in the different cloud scenarios.

For these simulations, we place emphasis on prescribing instrument performance metrics that are technologically feasible for a spaceborne radar deployment in the next decade, and will therefore provide an honest assessment of the platforms capabilities. In fact, only recently has it become possible to develop cloud radars at frequencies above 100 GHz with useful measurement





sensitivity for atmospheric science studies. The first in this new line of G-band cloud radars, the Jet Propulsion Laboratory's Vapor In-cloud Profiling Radar (VIPR) (Cooper et al., 2020; Roy et al., 2020, 2021), achieves a noise-equivalent reflectivity, a number which roughly dictates the minimum detectable cloud signal, of -40 dBZ at 1 km range using a range resolution of 15 m. If an identical system were deployed in low-earth orbit with an altitude of 400 km and range resolution of 200 m, the

resulting noise-equivalent reflectivity would become about 0 dBZ. This sensitivity level is 16 dB worse than the equivalent figure for CloudSat's 94 GHz Cloud Profiling Radar, and when combined with the increased attenuation at G-band in clear air and within clouds and precipitation, suggests that such a system would have minimal cloud sampling ability, especially in the PBL. Thus, significant improvements to the VIPR transceiver sensitivity are necessary for spaceborne implementation.

The radar performance metrics assumed here are consistent with a system that employs a long-duration pulse with linear

frequency modulation and pulse compression with very large time-bandwidth product, similar to that in the recent successful Radar in a CubeSat (RainCube) mission (Peral et al., 2019). We assume that the instrument is deployed on a traditional medium-sized satellite bus, similar in scale to the CloudSat platform, that can afford a large, solid reflector antenna of diameter $D_a = 2$ m, and a high-power, high-duty-cycle transmitter with a peak output power of 200 W. Such transmitter performance is consistent with a vacuum-electronics-based, traveling-wave tube amplifier that is currently under development, and which

is based on a heritage implementation at 231-235 GHz for a synthetic aperture radar application (Basten et al., 2016). The 3 transmit frequencies located at 155.5, 168.0, and 174.8 GHz are carefully chosen to lie in bands that do not feature international transmission restrictions (National Telecommunications & Information Administration (NTIA), 2015), and correspond to bands that are currently used in the VIPR system. Finally, while the desired vertical humidity profile resolution of 200 m suggests that the radar could have a range resolution of 200 m, we find that it is necessary to oversample the range dimension

by a small factor in order to remove humidity measurement bias that arises from non-uniform filling of range bins by cloud and precipitation particles. Therefore, we assume a range resolution of 50 m, which can be achieved with a moderate radar pulse modulation bandwidth of around 3 MHz.

## 2  Methods

### 2.1  LES case studies

The synthetic observations used in this study were produced by means of large-eddy simulation (LES), which is a well-known method for realistic high-resolution modeling of multi-phase three-dimensional non-hydrostatic turbulent atmospheric flows (Stevens and Lenschow, 2001). We choose five canonical cases covering the transition from the subtropics to the tropics over ocean. The simulated convective regimes were a focus of the previous field campaigns: DYCOMS-II and VOCALS (both on stratocumulus), BOMEX (non-precipitating shallow Cu), RICO (shallow Cu), and GATE (deep convection). Results of those

campaigns served as the initial and boundary conditions for the LES. Details of the field campaigns and the modeling setups are provided in 1 and the references therein.

For each simulation, the cascade of turbulent motions develops within the domain, with the boundary layer dynamics driven by surface fluxes of latent and sensible heat and the vertical shear of horizontal wind, modified by radiation (either interactive





**Table 1.** Summary of LES parameters for the 5 cases. *These ranges for GATE correspond to values at 24 hours and 6 hours into the simulation, respectively, with a monotonic decrease over this time window.

| Case Name | DYCOMS-II Second Dynamics and Chemistry of Marine Stratocumulus | VOCALS VAMOS Ocean-Cloud-Atmosphere-Land Study | BOMEX Barbados Oceanographic Meteorological Experiment | RICO Rain In Cumulus over the Ocean | GATE GARP Atlantic Tropical Experiment |
|---|---|---|---|---|---|
| PBL Type | Subtropical non-drizzling stratocumulus | Subtropical drizzling stratocumulus | Trade-wind nonprecipitating shallow convection | Trade-wind precipitating shallow convection | Tropical deep convection |
| Domain size (km x km) | 5 x 5 | 25.5 x 25.5 | 12.8 x 12.8 | 20.5 x 20.5 | 153.5 x 153.5 |
| Horizontal resolution (m) | 5 | 50 | 20 | 40 | 100 |
| Vertical resolution (m) | 5 | Variable, from 15m near the surface to 5m in the cloud layer | 20 | 40 | From 63m near the surface up to 300m near the tropopause |
| Microphysics | No precipitation | 2-moment (Morrison et al., 2005) | No precipitation | 2-moment (Morrison et al., 2005) | 1-moment (Lin et al., 1983) |
| Surface sensible heat flux (W/m2) | 15 | 9.6 | 9.5 | 7-8 | 15-18 |
| Surface latent heat flux (W/m2) | 115 | 188 | 153 | 155-170 | 97-106 |
| Liquid cloud fraction (%) | 98-100 | 98-100 | 16-19 | 20-24 | 30-52* |
| Ice cloud fraction(%) | 0 | 0 | 0 | 0 | 24-65* |
| Total cloud fraction (%) | 98-100 | 98-100 | 16-19 | 20-24 | 42-82* |
| TCWV (mm) | 22-23 | 13-14 | 35-36 | 42-43 | 51-52 |
| Surface precipitation (mm/h) | 0 | 0.002-0.008 | 0 | 0.02-0.07 | 0.6-0.7 |
| Case Reference | Stevens et al. (2005) | Berner et al. (2011) | Siebesma et al. (2003) | vanZanten et al. (2011) | Gentine et al. (2016) |
| Model | Matheou and Chung (2014) | Khairoutdinov and Randall (2003) | Matheou and Chung (2014) | Khairoutdinov and Randall (2003) | Skamarock et al. (2008) |

for stratocumulus or prescribed for other cases) and large-scale advective tendencies (for shallow Cu cases). For the shallow precipitating cases, 2-moment microphysics provides the information about both mixing ratios and droplet number concentrations of the precipitating and non-precipitating forms of water. For deep convection, bulk 1-moment microphysics is used that represents six classes of water: vapor, cloud liquid, cloud ice, warm rain, snow, and graupel. Horizontal resolutions range from

5  5 m for DYCOMS-II to 100 m for GATE. Three-dimensional LES outputs are combined with one-dimensional MERRA-2 reanalysis data aloft producing a full-depth atmospheric column that is the input for the radiative transfer model.

## 2.2 Radar forward model

We implement the radar forward model on a given LES atmospheric state in three distinct steps: (1) calculation of volumetric and surface scattering parameters for each of the DAR channel frequencies at the LES model resolution; (2) interpolation in

10  the vertical dimension to an equally spaced grid that is much finer than the radar range resolution, and subsequent computation of the time-dependent radiative transfer solution including the effects of multiple scattering; and (3) processing of the ideal, high-resolution radar backscatter quantities to produce observations consistent with the assumed radar instrument parameters. These steps are outlined in the following three subsections.



### 2.2.1 Single-scattering properties

*Atmospheric gases* – At millimeter-wave frequencies in the troposphere, the only relevant beam interaction with atmospheric gases is absorption by molecular oxygen and water vapor. We calculate the absorption properties using the millimeter-wave propagation model from the Earth Observing System Microwave Limb Sounder, which includes both line-by-line and continuum absorption contributions for each species (Read et al., 2004; Read et al., 2006).

5 *Liquid hydrometeors* – We treat the two LES liquid hydrometeor species, cloud and rain, as dielectric spheres and calculate the single-scattering properties using Mie scattering theory (Bohren and Huffman, 2004). The dielectric constant of pure water is calculated using the parameterization described in Liebe et al. (1991), which is an updated version of the classic model by Ray (1972). The particle size distributions (PSDs) for all hydrometeor species are parameterized using the modified gamma distribution,

$$N(D) = \frac{N_0}{\Gamma(\nu)} \left( \frac{D}{D_n} \right)^{\nu-1} \frac{1}{D_n} e^{-D/D_n}, \tag{1}$$

where $N_0$ is the number concentration with units of particle number per unit volume, $\nu$ is the distribution shape parameter which can vary with species, and $D_n$ is a characteristic diameter. Because the LES models used in this work include both single-moment and double-moment microphysics schemes, our radar forward simulator features two different parameterizations for calculating volume scattering quantities by integrating the single-particle, single-scattering quantities over a given PSD. For the 2-moment schemes, the prognostic species number concentration and mass mixing ratio uniquely determine $D_n$ once $\nu$ is prescribed, and therefore permit straightforward calculation of the PSD-integrated quantities. For models employing 1-moment schemes, however, an additional microphysical constraint is needed. In this work, all such constraints take on the form

$$N_0 = x_1 D_n^{x_2} e^{-x_3(T-T_0)}, \tag{2}$$

where the parameters $x_1$, $x_2$, and $x_3$ have been determined from studies of comprehensive in-situ microphysical observations, and $T_0 = 273.15$ K. All particle species except for cloud have 1-moment parameterizations that assume an exponential PSD, or $\nu = 1$. For the cloud species, we assume a constant $N_0$ that is consistent with observations (Miles et al., 2000) and use the same shape parameter as for the 2-moment scheme. The shape parameters and parameterization details for the cloud and rain species are detailed in Table 2.

*Ice hydrometeors* – While the majority of convective regimes studied in this work include liquid-phase particles only, the case of deep convection necessitates modeling of radar scattering from three species of ice-phase particles: ice, snow, and graupel. Unlike cloud and rain droplets, the ice hydrometeors feature non-trivial relationships between particle mass $m$ and maximum linear dimension $D$, so-called mass-dimenisonal relations:

$$m(D) = a_m D^{b_m}. \tag{3}$$

The mass-dimensional relation and PSD shape parameters for the 2-moment implementation described here are taken from the Regional Atmospheric Modeling System (RAMS) 2-moment microphysics parameterization (Walko et al., 1995; Meyers



**Table 2.** *The dendrite growth model uses the Leinonen and Szyrmer (2015) implementation of the algorithm presented in Reiter (2005).

| Species | $a_m$ $(\mathrm{kg\,m}^{-b_m})$ | $b_m$ | Particle Shape | $\nu$ (2-Mom.) | $\nu$ (1-Mom.) | $x_1$ $(\mathrm{m}^{-3-x_2})$ | $x_2$ | $x_3$ $(\mathrm{K}^{-1})$ | 1-Moment Reference |
|---|---|---|---|---|---|---|---|---|---|
| Cloud | 524 | 3 | Sphere | 4 | 4 | $7.4 \times 10^7$ | 0 | 0 | Miles et al. (2000) |
| Rain | 524 | 3 | Sphere | 2 | 1 | 26 | -0.57 | 0 | Abel and Boutle (2012) |
| Ice | 110.8 | 2.91 | Hex. column | 2 | 1 | $2.0 \times 10^6$ | 1 | 0.12 | Cox (1988) |
| Snow | $2.74 \times 10^{-3}$ | 1.74 | Dendrite* | 2 | 1 | $1.4 \times 10^4$ | 1 | $7.2 \times 10^{-2}$ | Wood (2011) |
| Graupel | 157 | 3 | Sphere | 2 | 1 | $7.9 \times 10^9$ | -3.58 | 0 | Field et al. (2019) |

et al., 1997; Tatarevic et al., 2019), and are presented in Table 2. While this implies that the microphysical parameterizations in the radar simulator differ from those in the LES models, this difference should have little impact on the fidelity of the forward simulated observations, since the LES models provide only bulk parameters describing the PSD.

Before calculating single-particle scattering quantities, we must specify details of the three-dimensional shape and composition of the ice species. For graupel, we assume a homogeneous spherical geometry with reduced density relative to solid 5 ice that is determined by the mass-dimensional relation, and calculate effective dielectric and scattering properties using the Maxwell-Garnet effective medium approximation and Mie scattering theory, respectively. For the ice and snow species, we assume particle shapes of hexagonal columns and dendrites, respectively, and generate discretized, three-dimensional crystal models using the approach of Leinonen and Szyrmer (2015). In this case, the mass-dimensional relations given in Table 2 provide the necessary constraint to fix the size-dependent aspect ratios of the columnar and dendritic crystals. We then calculate 10 the scattering properties for ice and snow using the discrete dipole approximation (DDA) as implemented in the Amsterdam DDA (ADDA) solver of Yurkin and Hoekstra (2011). Since we expect the DDA results to converge to those of the T-matrix method (Mishchenko et al., 1996) for sufficiently small particle size, we specify an alternative bulk representation of the ice and snow species as homogeneous cylinders and spheroids with effective dielectric properties calculated again using the Maxwell-Garnet approximation, and with the same size-dependence of the aspect ratio as the 3D models. T-matrix calculations are then performed using the solver of Mishchenko and Travis (1998) as implemented in the Python package "PyTMatrix" by Leinonen (2014). Once convergence of DDA to T-matrix is established at some minimum particle size, the T-matrix values are used for all subsequent smaller particle sizes. For all ice hydrometeor species, the frequency-dependent dielectric constant values of solid ice are taken from Warren and Brandt (2008).

Comparisons of the size-dependent backscattering cross section calculated using the Mie, T-matrix, and DDA methods for ice and snow are shown in Fig. 1. We note that DDA calculations for a particular particle size are performed for a single orientation that is chosen randomly for each particle realization, and therefore do not correspond to orientation-averaged quantities. Instead, it is assumed that the integration over the PSD to derive volume scattering coefficients provides sufficient averaging over particle orientation.

*Ocean surface* – Scattering from the ocean surface is treated within the geometrical optics (GO) approximation, in which backscatter is dominated by quasi-specular reflection from normally oriented wave facets (Kodis, 1966; Barrick, 1968; Valenzuela, 1978). At nadir incidence within the GO approximation, the ocean surface normalized radar cross section (NRCS) $\sigma^0$




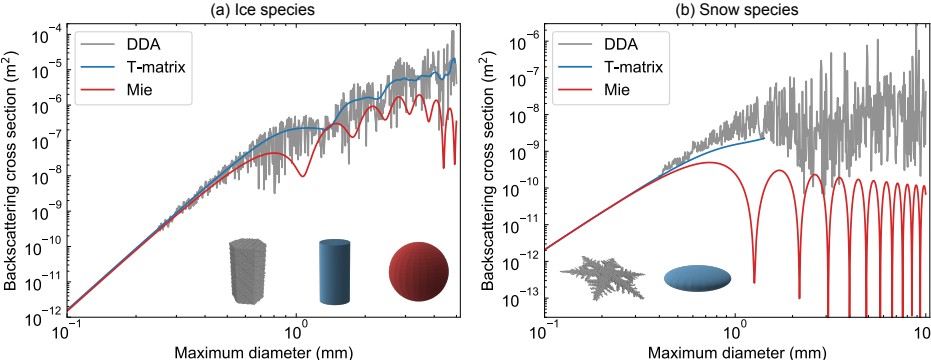

**Figure 1.** Comparison of DDA, T-matrix, and Mie backscattering cross section calculations for the two solid-phase hydrometeor species described in this work. The ice (a) and snow (b) species are treated as hexagonal columns and pristine dendrites for DDA calculations and as cylinders and oblate spheroids for the T-matrix approach, respectively. For the snow T-matrix calculations, a maximum diameter greater than 1.5 mm results in a spheroid aspect ratio that is beyond the convergence criteria for the T-matrix routine.

depends only on the mean square slopes of the ocean surface in the upwind and crosswind directions and the transmit frequency via the sea water dielectric constant. Here we use the classic parameterization of Cox and Munk (1954) to express the mean

square slopes in terms of the near-surface wind, and the millimeter-wave dielectric constant model of Meissner and Wentz (2004), for which we assume a constant salinity of 35 ppt. Lastly, we apply a fixed offset of +1.5 dB to the GO prediction for all frequencies and wind speeds in order to match recent airborne observations of $\sigma^0$ at low wind speeds using the VIPR instrument (Roy et al., 2021).

### 2.2.2   Radiative transfer

The single-scattering properties of the atmospheric gases, cloud and precipitation hydrometeors, and ocean surface at the LES spatial resolution then serve as inputs to a multiple scattering simulator, for which we utilize the time-dependent two-stream implementation of Hogan and Battaglia (2008). In order to assess systematic error resulting from non-uniform range-bin filling, we interpolate the scattering property fields from the LES grid resolution — which has a variable spacing that increases with height — to an equally spaced vertical grid with 10 m resolution. This ensures that the instrument range resolution of $\Delta r = 50$

m is sufficiently oversampled in the forward simulator throughout the LES domain. At this stage, the only instrument parameter that is required to be specified for the multiple scattering calculation is the angular width of the radar beam, which we calculate from the antenna diameter $D_a$ assuming a Gaussian beam shape and an 11 dB beam taper. The output of the multiple scattering calculation is the observed, or apparent, volume backscatter coefficient $\eta_{\text{obs}}(f, x, y, z)$ as a function of transmit frequency at the LES horizontal resolution $(\Delta x, \Delta y)$ and interpolated vertical resolution (10 m).



### 2.2.3 Simulated observations

Given the forward-simulated, observed, volume backscatter coefficient, $\eta_{\mathrm{obs}}(f,x,y,z)$, we first average over the instrument horizontal beam footprint and range resolution and assume that the along-track direction corresponds to the $y$ dimension. Note that in this step of resolution degradation we are leveraging the fine spatial resolution of LES to provide realistic representation of non-uniform beam filling effects both in terms of the horizontal footprint and within single range bins. Given an antenna footprint $|F(f,\theta,\phi)|^2 = |F(f,\boldsymbol{r})|^2$, which we express in the interpolated Cartesian coordinate system, we calculate the footprint-averaged, observed backscatter after an along-track integration time of $T_{\mathrm{int}}$ from

$$\bar{\eta}_{\mathrm{obs}}(f,x_i,y_j,z_k) = \frac{1}{\Delta r R^2 \Omega_{\mathrm{eff}} T_{\mathrm{int}}} \int\limits_{V_{ijk}} d^3\boldsymbol{r}' \eta_{\mathrm{obs}}(f,\boldsymbol{r}') \int\limits_{0}^{T_{\mathrm{int}}} dt' |F(f,x',y'-v_g t')|^4. \tag{4}$$

Here the orbital altitude $R$ appears via the solid-angle normalization factor, where we've substituted $d\Omega' = dx'dy'(R-z_k)^{-2} \approx dx'dy'R^{-2}$ throughout the LES domain for simplicity since the true range from the satellite $R-z_k$ varies by only a few percent. The integration volume $V_{ijk}$ centered on the observation point $(x_i,y_j,z_k)$ is defined by $-\Delta x_{\mathrm{3dB}}/2 \leq x'-x_i \leq \Delta x_{\mathrm{3dB}}/2$, $-\Delta y_{\mathrm{3dB}}(T_{\mathrm{int}})/2 \leq y'-y_j \leq \Delta y_{\mathrm{3dB}}(T_{\mathrm{int}})/2$, and $-\Delta r/2 \leq z'-z_k \leq \Delta r/2$, where $\Delta x_{\mathrm{3dB}}$ is the 3 dB full width of the two-way beam pattern $|F(f_0,\boldsymbol{r})|^4$ at the lowest frequency, and $\Delta y_{3dB}(T_{\mathrm{int}})$ is the 3 dB full width of the time-averaged beam footprint in the along-track direction. Finally, the effective two-way solid angle $\Omega_{\mathrm{eff}}$ is the integral of the time-averaged two-way beam pattern over the restricted angular domain defined by $V_{ijk}$. Averaging the observed NRCS over the satellite footprint to get $\bar{\sigma}^0_{\mathrm{obs}}(f,x_i,y_j)$ is performed by making the replacement $\eta_{\mathrm{obs}}(f,\boldsymbol{r}') \rightarrow \Delta r \delta(z') \sigma^0_{\mathrm{obs}}(f,x',y')$ in Eq. (4), where $\delta(z')$ is the Dirac delta function.

We evaluate Eq. (4) for each footprint centroid $(x_i,y_j)$ by performing a discrete summation over the relevant variables at the interpolated, model resolution. The radar sampling horizontal grid is defined by $x_{i+1}-x_i = \Delta x_{\mathrm{3dB}}$ and $y_{i+1}-y_i = v_g T_{\mathrm{int}}$, implying that adjacent footprints in the cross-track dimension contain no common LES pixels, while sequential along-track footprints feature realistic overlap. Note that we do not assume that the radar can measure multiple, simultaneous footprints in the cross-track dimension, but instead are maximizing use of the rectangular LES domain to gather as many statistical samples as possible for eventual retrieval performance evaluation. Finally, for comparison of radar retrieved humidity profiles with those in the LES, we average the model humidity field over the time-averaged, two-way antenna pattern as well.

Next, we calculate the observed reflectivity from $Z_{\mathrm{obs}}(f) = \bar{\eta}_{\mathrm{obs}}(f)c^4 f^{-4}\pi^{-5}|K_w(f)|^{-2}$, where $c$ is the speed of light, $K_w(f) = [\varepsilon_w(f)+2]^{-1}[\varepsilon_w(f)-1]$, and $\varepsilon_w(f)$ is the dielectric constant of pure water evaluated at 280 K. At this point we must specify the instrument parameters that determine the minimum detectable signal and random measurement uncertainty. These values are listed in Table 3. As previously discussed, we assume a transmitter with a pulse-averaged power of 200 W and duty cycle of 25%. Then, in order to determine the single-pulse radar sensitivity, we must specify the pulse duration $\tau_p$ and compare this with the measurement decoherence time. Generally speaking for a pulse compression radar that is not performing Doppler velocity estimation, it is desirable to have $\tau_p$ as large as possible while maintaining *coherent* measurement integration to maximize signal-to-noise ratio (SNR). However, for a spaceborne radar in low-earth orbit, the fast transit of the beam across a region of interest combined with the fact that the measurement relative error is at minimum $\sigma_Z Z_{\mathrm{obs}}^{-1} = N_i^{-1/2}$, where $N_i$



**Table 3.** High-level airborne VIPR system parameters.

| Radar Parameter | Value | | |
|---|---|---|---|
| Transmit power, $P_t$ | 200 W | | |
| Transmitter duty cycle | 25% | | |
| Pulse duration, $\tau_p$ | 50 $\mu$s | | |
| Pulse repetition interval, $T_{\text{rep}}$ | 200 $\mu$s | | |
| Range resolution, $\Delta r$ | 50 m | | |
| Antenna diameter, $D_a$ | 2 m | | |
| Receiver noise figure | 8 dB | | |
| Orbital altitude, $R$ | 400 km | | |
| Orbital speed, $v_{\text{sat}}$ | 7.7 km s$^{-1}$ | | |
| Ground speed, $v_g$ | 7.2 km s$^{-1}$ | | |
| Along-track integration time, $T_{\text{int}}$ | 60 ms | | |
| Number of pulses per frequency, $N_p$ | 100 | | |
| Transmit frequency | $f_0 = 155.5$ GHz | $f_1 = 168.0$ GHz | $f_2 = 174.8$ GHz |
| 3 dB beam width | 0.064° | 0.060° | 0.057° |
| Horizontal footprint* ($x \times y$) | 450 × 640 m | 420 × 620 m | 400 × 610 m |
| Minimum detectable reflectivity, dB$Z_{\text{min}}$ | -33 dBZ | -34 dBZ | -35 dBZ |

is the number of statistically independent pulses transmitted at a given frequency, implies that $\tau_p$ should not be so large that insufficient *incoherent* averaging occurs.

Because we expect measurement decorrelation to be dominated in general by broadening of the returned signal spectrum due to the satellite-motion-induced Doppler effect, we can quantify the decoherence time scale by analyzing the resulting pulse-to-pulse correlations. The relevant equations are discussed in several references (Tanelli et al., 2002; Hogan et al., 2005) and are revisited in Appendix A, with the end result that the pulse-to-pulse decoherence time scale for radar signal power measurements, sometimes referred to as the "time to independence", is given by $\tau_i = (2\sqrt{\pi}\sigma_f)^{-1}$, where $\sigma_f = v_{\text{sat}}\theta_0\lambda^{-1}$ is

the standard deviation of the Doppler-effected spectral density for a stationary target that fills the radar beam. However, because of the inherent scaling $\theta_0 \propto \lambda/D_a$, the Doppler spectral width in frequency space is independent of the radar frequency, and for our assumed hardware parameters the coherence time becomes $\tau_i \approx 0.40 D v_{\text{sat}}^{-1} = 104$ $\mu$s. Given $N_p$ sequential pulses with an inter-pulse spacing of $T_p$, the number of statistically independent pulses for radar signal power estimation $N_i$ is determined from (Doviak and Zrnić, 1993)

$$\xi(\tau_i, T_p) \equiv \frac{N_p}{N_i} = 1 + 2\sum_{m=1}^{N_p-1} \left(1 - \frac{m}{N_p}\right) \exp\left(-\frac{m^2 T_p^2}{\tau_i^2}\right). \tag{5}$$

Because the radar cycles between widely separated transmit frequencies after each pulse, the relevant inter-pulse spacing in this case is $T_p = N_f T_{\text{rep}}$, where $N_f = 3$ is the number of frequency channels. Evaluating Eq. (5) using the parameters in Table 3, we find that $\xi \approx 1$, or $N_p = N_i$. Next, we calculate the relative uncertainty in the reflectivity measurement due to random error according to (Papoulis, 1965; Doviak and Zrnić, 1993; Torres, 2001)

$$\frac{\sigma_Z}{Z_{\text{obs}}} = \frac{1}{\sqrt{N_p}} \left[\xi(\tau_i, T_p) + \frac{2}{\text{SNR}} + \frac{1}{\text{SNR}^2}\right]^{1/2}, \tag{6}$$





where SNR is the measurement signal-to-noise ratio, which is easily determined in our simulations by comparing the calculated radar echo power with the prescribed receiver noise $P_n = k_B T_B F \tau_p^{-1}$. Here $k_B$ is Boltzmann's constant, $T_B$ is the scene brightness temperature, assumed to be 280 K, and $F$ is the noise factor related to the noise figure through $F = 10^{NF/10}$. Note that Eq. (6) assumes that the noise power contribution to the detected signal can be removed without inflating the measurement variance. The noise-equivalent reflectivity $Z_{\mathrm{NE}}$ and surface NRCS $\sigma_{\mathrm{NE}}^0$ are defined as the respective observed values that correspond to SNR $= 1$, while the minimum detectable reflectivity and NRCS improve upon this figure as a result of incoherent

averaging according to $\mathrm{dB}Z_{\min} = \mathrm{dB}Z_{\mathrm{NE}} - 10\log_{10}(\sqrt{N_i})$ and $\sigma_{\min}^0(\mathrm{dB}) = \sigma_{\mathrm{NE}}^0(\mathrm{dB}) - 10\log_{10}(\sqrt{N_i})$.

## 2.3 DAR measurement methodology

The physical basis of the DAR measurement has been discussed in a number of previous works. Here, we focus on the complete treatment first detailed in Battaglia and Kollias (2019) and subsequently in Roy et al. (2020) that quantifies the impact of frequency-dependent hydrometeor scattering and attenuation on the DAR humidity retrieval, and simply recount the main

results. First, we define the local, observed extinction coefficient,

$$\beta_{\mathrm{obs}}(r, f) = \frac{1}{2}\frac{\partial}{\partial r}\ln\left(Z_{\mathrm{obs}}(r, f)\right). \tag{7}$$

Anticipating that differential extinction will be measured over a baseline of $N_b$ range bins, we form the finite difference version of Eq. (7),

$$\beta_{\mathrm{obs}}(r, f) \approx \frac{1}{2N_b\Delta r}\ln\left(\frac{Z_{\mathrm{obs}}(r^-, f)}{Z_{\mathrm{obs}}(r^+, f)}\right), \tag{8}$$

where $r^{\pm} = r \pm N_b\Delta r/2$. For small changes in range $N_b\Delta r$ and assuming negligible multiple scattering, the appropriate fitting function for $\beta_{\mathrm{obs}}$ can be shown to be (Battaglia and Kollias, 2019; Roy et al., 2020)

$$\hat{\beta}_{\mathrm{obs}}(r, f_i) = a_1 + a_2(f_i - f_0) + a_3\kappa_v(r, f_i) + \beta_{g,d}(r, f_i), \tag{9}$$

where we assume that measurements are made at a discrete set of frequencies $\{f_i\}$, the values $a_j$ are the regression coefficients, $\kappa_v(r, f)$ is the water vapor absorption cross section per unit mass, and $\beta_{g,d}(r, f)$ is the absorption coefficient for dry air.

Defining the unattenuated reflectivity $Z(r, f) = Z_{\mathrm{obs}}(r, f)\exp(2\tau(r, f))$, where $\tau$ is the one-way optical depth, the differential backscatter function $\alpha(r, f) = (2N_b\Delta r)^{-1}\ln[Z(r^-, f)/Z(r^+, f)]$, and the hydrometeor extinction coefficient $\beta_h(r, f)$, the regression coefficients in Eq. 9 can be shown to relate to the following physical quantities:

$$a_1 \leftrightarrow \alpha(r, f_0) + \beta_h(r, f_0)$$

$$a_2 \leftrightarrow \frac{\partial}{\partial f}\left(\alpha(r, f) + \beta_h(r, f)\right)|_{f=f_0}$$

$$a_3 \leftrightarrow \rho_v(r). \tag{10}$$

In these relations, it is understood that all functions of $r$ are approximated by their average value between $r^-$ and $r^+$.

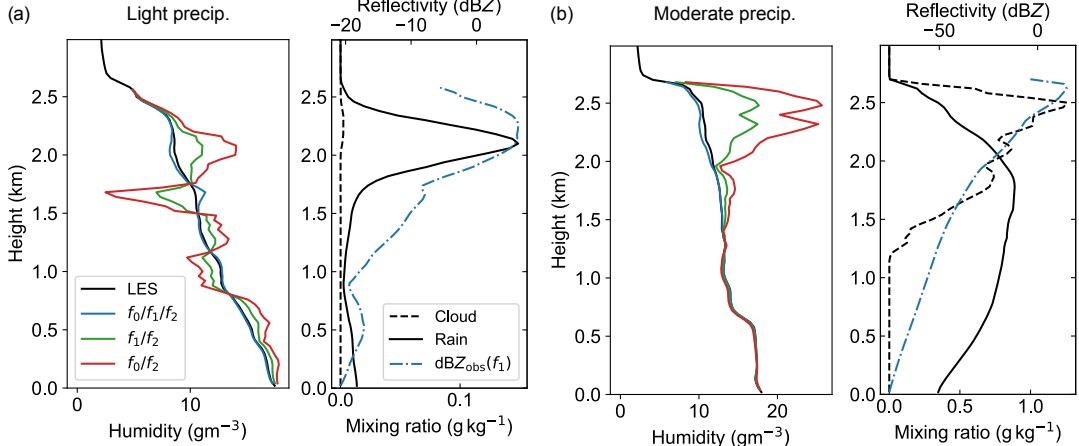

**Figure 2.** Comparison of 2-frequency and 3-frequency DAR humidity retrievals using atmospheric profiles from the RICO LES. An ideal reflectivity measurement is assumed, with no multiple scattering, arbitrarily high range resolution, and zero random measurement error. Retrievals are performed using a differential absorption baseline equal to the model vertical reoslution of $r^+ - r^- = 40$ m. (a) Humidity retrievals vs. LES truth (left panel) for a lightly precipitating grid cell. (Right panel) Liquid water mixing ratio profiles and corresponding observed reflecitvity at $f_1 = 168$ GHz. (b) Same as (a) for a moderate precipitation case. In both cases, performing the 3-parameter fit using the model function in Eq. (9) removes the humidity biases that originate from frequency dependence of hydrometeor backscatter and extinction.

With 3 regression parameters, the model in Eq. (9) requires observations at 3 or more frequencies in order to perform a least-squares fit. For $N_f = 3$, the solution is independent of the measurement covariance matrix and can be written as $\hat{x} = K^{-1}(y - b)$, where $\hat{x} = [\hat{a}_1, \hat{a}_2, \hat{a}_2]^T$, $K_{i0} = 1$, $K_{i1} = f_i - f_0$, $K_{i2} = \kappa_v(r, f_i)$, $b_i = \beta_{g,d}(r, f_i)$, and $y_i = \beta_{obs}(r, f_i)$. It is instructive to use an example LES hydrometeor profile to compare humidity profiles estimated using this 3-frequency model and those derived using the 2-frequency differential absorption approach (e.g., Lebsock et al. (2015); Millán et al. (2016); Roy et al. (2018)). Specifically, in the 2-frequency approach, one assumes that the contributions from $a_2$ and $\beta_{g,d}$ to observed differential extinction in Eq. (9) are negligible, and therefore can estimate the humidity as $\hat{\rho}_v(r) = [\beta_{obs}(r, f_1) - \beta_{obs}(r, f_0)][\kappa_v(r, f_1) - \kappa_v(r, f_0)]^{-1}$. Fig. 2 compares the humidity profiles retrieved using these two approaches for two different atmospheric profiles from the RICO LES case with different precipitation characteristics. To highlight the fact that the 2-frequency DAR measurement can exhibit biased retrievals even for a perfect radar measurement, here we use reflectivity profiles calculated within the single-scattering approximation at the LES model resolution (i.e., not averaged over the radar footprint). These results show the added utility of a third DAR frequency channel over a wide transmitter bandwidth to eliminate hydrometeor scattering induced biases in the humidity retrieval.





## 2.4 Water vapor retrieval algorithm

The retrieval algorithm employed in this work follows a similar formalism as that developed in Roy et al. (2020), but with some improvements and important modifications to accommodate the $N_f \geq 3$ DAR approach and additional regression parameter at each retrieval height corresponding to the $a_2$ term that is linear in frequency in Eq. (9). In essence, the task is to relate the radar observations $\boldsymbol{y}$ to a forward model $\boldsymbol{F}(\boldsymbol{x},\boldsymbol{c})$ that is a generalization of the fitting function in Eq. (9) to model the entire vertical profile. Here $\boldsymbol{x}$ is the state vector consisting of quantities that will be retrieved as part of the inverse problem and $\boldsymbol{c}$ contains parameters that must be assumed to compute the forward model. We note at the outset that the inverse problem will be formulated in such a way that the forward model will be a linear transformation of the state vector, or $\boldsymbol{F} = K\boldsymbol{x} + \boldsymbol{b}$, where $\boldsymbol{b}$ is the part of the forward model that does not depend on the state vector, and it is understood that $K$ and $\boldsymbol{b}$ depend on the forward model assumptions $\boldsymbol{c}$.

We define the observation vector as $\boldsymbol{y} = [\boldsymbol{y}_0, \boldsymbol{y}_1, \boldsymbol{y}_2]$, where $[\boldsymbol{y}_j]_i = \ln(Z_{\mathrm{obs}}(r_i, f_j)Z_0^{-1})$ for $0 \leq i \leq M-2$, $[\boldsymbol{y}_j]_{M-1} = \ln(\sigma_{\mathrm{obs}}^0(f_j))$, and $\boldsymbol{r} = [r_0, r_1, \ldots, r_{M-1}]$ contains the range bin positions for which the radar signals exceed the instrument sensitivity threshold *at all 3 frequencies*. Given the full range vector $\boldsymbol{r}' = [r_{\mathrm{TOA}}, r_{\mathrm{TOA}} + \Delta r, ..., r_s - \Delta r, r_s]$ of length $M'$ where $r_{\mathrm{TOA}}$ and $r_s$ are the top-of-atmosphere and surface ranges, respectively, we define the measurement projection matrix $P_r$ according to $\boldsymbol{r} = P_r \boldsymbol{r}'$. In cases of severe hydrometeor beam attenuation, the surface return may be below the sensitivity threshold $\sigma_{\mathrm{min}}^0$, in which case the final element of each $\boldsymbol{y}_j$ is replaced by the last cloud bin with $\mathrm{dB}Z_{\mathrm{obs}} > \mathrm{dB}Z_{\mathrm{min}}$. The measurement covariance matrix is defined in block-diagonal form as $S_y = \mathrm{diag}(S_{y_0}, S_{y_1}, S_{y_2})$, where $[S_{y_j}]_{ik} = (\sigma_Z Z_{\mathrm{obs}}^{-1}|_{r_i, f_j})^2 \delta_{ik}$, and the relative error $\sigma_Z Z_{\mathrm{obs}}^{-1}$ is evaluated at range $r_i$ and frequency $f_j$ using Eq. (6). Note that in general $\boldsymbol{y}$ has dimension $N_f \times M$ and that this formalism is easily extended to more than 3 frequencies.

Next, we define the state vector according to $\boldsymbol{x} = [\boldsymbol{x}_0, \boldsymbol{x}_1, \boldsymbol{x}_2]$, where $[\boldsymbol{x}_0]_i = \ln(Z_r(r_i, f_0)Z_0^{-1})$ is a vector of dimension $M$, $Z_r(r, f_0)$ is the retrieved reflectivity at $f_0$ including attenuation from hydrometeors, $\boldsymbol{x}_1 = [\gamma_0, \gamma_1, \ldots, \gamma_{M-1}]$ where $\gamma_i$ is a range-dependent slope vs. frequency and is analogous to $a_2$ in Eq. (9), and $\boldsymbol{x}_2$ contains the $N$ retrieved humidity values and requires careful specification given an arbitrary measurement projection $P_r$. The forward model is then defined in block form as follows:

$$\boldsymbol{F}(\boldsymbol{x},\boldsymbol{c}) = \begin{pmatrix} I_M & (f_0 - f_0)I_M & -2T_0 \\ I_M & (f_1 - f_0)I_M & -2T_1 \\ I_M & (f_2 - f_0)I_M & -2T_2 \end{pmatrix} \begin{pmatrix} \boldsymbol{x}_0 \\ \boldsymbol{x}_1 \\ \boldsymbol{x}_2 \end{pmatrix} - 2 \begin{pmatrix} \boldsymbol{\tau}_{g,d}(f_0) \\ \boldsymbol{\tau}_{g,d}(f_1) \\ \boldsymbol{\tau}_{g,d}(f_2) \end{pmatrix}, \tag{11}$$

where $I_M$ is the $M \times M$ identity matrix, $T_i$ is a matrix of size $M \times N$ that computes the water vapor absorption contribution to the total optical depth profile given the values in $\boldsymbol{x}_2$, and $\boldsymbol{\tau}_{g,d}(f_i)$ is the optical depth profile at $f_i$ due to dry air absorption. At this point, we must assume vertical profiles of temperature $T(r')$ and pressure $P(r')$ at the radar range resolution, which are elements of $\boldsymbol{c}$, in order to calculate the gas absorption quantities $\boldsymbol{\tau}_{g,d}(f_i)$ and $T_i$. For this work, we do so by averaging these thermodynamic profiles over the whole LES domain, and smoothing the result by performing a convolution with a box of length 2 km. Note that due to the relatively weak dependence of the water vapor absorption cross section on pressure and temperature, the systematic error associated with these assumptions is not a leading order effect (Roy et al., 2018). Then, the





dry air optical depth vector is easily defined by the recursive relation $[\boldsymbol{\tau}_{g,d}(f_j)]_i = [\boldsymbol{\tau}_{g,d}(f_j)]_{i-1} + \Delta r \beta_{g,d}(r'_{i-1}, f_j)$ and the initial condition $[\boldsymbol{\tau}_{g,d}(f_j)]_0 = 0$.

The first step in defining $T_i$, and therefore the structure and interpretation of $\boldsymbol{x}_2$, is to prescribe a maximum vertical resolution for the retrieved humidity profile, which we call $\Delta z$. Furthermore, we restrict $\Delta z$ to values that satisfy the condition that

$\Delta z / \Delta r = O$ is an integer, and define the new vertical axis $\boldsymbol{z}' = [z_{\text{TOA}}, z_{\text{TOA}} - \Delta z, \ldots, \Delta z + \Delta r, \Delta r]$, where the lowest height position corresponds to the first range bin above the surface $r_s - \Delta r$, and $z_{\text{TOA}}$ corresponds to the range $r' = r_{\text{TOA}} + \Delta z - \Delta r$. In this work, we use $\Delta z = 200$ m, resulting in a range bin oversampling factor of $O = 4$. To determine which vertical humidity values will be retrieved given a radar measurement projection $P_r$, we define a new projection matrix $P_z$ and axis $\boldsymbol{z}$, with $\boldsymbol{z} = P_z \boldsymbol{z}'$, and $P_z$ projects onto the non-zero elements of the following downsampling vector $\boldsymbol{d}$:

$$[\boldsymbol{d}]_i = \sum_{j=0}^{M-1} \sum_{k=Oi}^{Oi+O-1} [P_r]_{jk}. \tag{12}$$

Note that this definition for $\boldsymbol{z}$ ensures that a humidity value at a certain height $z$ will only be retrieved if there is at least one radar observation within the interval $[z - \Delta r, z + (O-1)\Delta r)$.

Next, we write the matrix $T_i$ as the product of three matrices, $T_i = P_r T'_i A$, where $A$ $(M' \times N)$ first interpolates the humidity state vector $\boldsymbol{x}_2$ to the $\boldsymbol{r}'$ full range space, then $T'_i$ $(M' \times M')$ computes the full optical depth profile, and finally $P_r$ projects this profile onto the measurement range vector $\boldsymbol{r}$. In contrast with the humidity interpolation approach utilized in Roy et al.

(2020), here we interpolate to the $\Delta r$ resolution between retrieval heights in $\boldsymbol{z}$ using an exponential profile with fixed scale height $H_\rho$. For situations where sequential elements for $\boldsymbol{z}$ are closely spaced, the choice of interpolation function is irrelevant, since the absorption line shape properties change negligibly and thus the retrieval is only sensitive to the integrated water vapor (IWV) between the two retrieval heights. However, for situations with widely spaced elements of $\boldsymbol{z}$, including TCWV retrieval scenarios where no clouds are detected, the exponentially interpolated humidity profile provides a realistic weighting of the

vertical absorption cross section. Finally, the matrix $T'_i$ can be shown to be a lower triangular matrix defined by:

$$T'_i = \Delta r \begin{pmatrix} \kappa_v(r'_0, f_i) & 0 & \cdots & & & 0 \\ \kappa_v(r'_0, f_i) & \kappa_v(r'_1, f_i) & 0 & \cdots & & \vdots \\ \kappa_v(r'_0, f_i) & \kappa_v(r'_1, f_i) & \kappa_v(r'_2, f_i) & 0 & \cdots & \vdots \\ \vdots & \vdots & \vdots & \ddots & \ddots & \vdots \\ & & & & \ddots & 0 \\ \kappa_v(r'_0, f_i) & \kappa_v(r'_1, f_i) & \kappa_v(r'_2, f_i) & \cdots & \cdots & \kappa_v(r'_{M'-1}, f_i) \end{pmatrix} \tag{13}$$

An example of the humidity profile grid selection and interpolation procedure is given in Appendix B.

To proceed towards retrieving the best estimate of the state vector $\boldsymbol{x}$, we define the cost function to equal the weighted least-squares sum $C(\boldsymbol{x}) = (\boldsymbol{y} - K\boldsymbol{x} - \boldsymbol{b})^T S_y^{-1} (\boldsymbol{y} - K\boldsymbol{x} - \boldsymbol{b})$. Note that, because of the introduction of the model parameters in

$\boldsymbol{x}_1$ corresponding to the linear frequency fit term, the DAR retrieval no longer requires regularization as was implemented in Roy et al. (2020), and there is no need to introduce a systematic error covariance matrix, because the current model is designed





to eliminate the hydrometeor-scattering-induced bias. Furthermore, it worth highlighting here that the only a priori information necessary for this retrieval is specification of representative $T$ and $P$ profiles, which exemplifies the sharp contrast between this inversion approach and that of optimal estimation. The value of $\boldsymbol{x}$ that minimizes $C(\boldsymbol{x})$, as well as the estimated state's covariance matrix, are then calculated as follows:

$$\hat{S}_x = (K^T S_y^{-1} K)^{-1}$$
$$\hat{\boldsymbol{x}} = \hat{S}_x K^T S_y^{-1} (\boldsymbol{y} - \boldsymbol{b}). \tag{14}$$

For much of the analysis in Section 3, it is more appropriate to analyze the IWV for individual partial columns of the retrieval (i.e., corresponding to a single column of the matrix $A$ as shown in Fig. B1(b)) than it is water vapor densities at specific heights, as this provides a common framework to discuss both in-cloud profiling retrievals with high vertical resolution and partial and total column retrievals encompassing large cloud-free volumes. It is easy to show that converting the estimated humidity state vector $\boldsymbol{x}_2$ to a partial-column IWV variable amounts to a simple linear transformation $\boldsymbol{c} = L\boldsymbol{x}_2$, where $L_{ii} = \Delta r \sum_j A_{ji}$ is a diagonal matrix where each entry is the sum of all elements in a column of $A$. The corresponding IWV covariance matrix is then calculated according to $S_c = L S_{x_2} L^T$, where $S_{x_2}$ is the square block from $\hat{S}_x$ that pertains to the retrieved humidity values $\boldsymbol{x}_2$.

## 3 Results

In this section we detail the results of case-by-case forward simulations and retrievals for the 5 LES scenarios described in Section 2.1. Because the analysis framework is identical for all 5 cases, we provide the most detail in Section 3.1 in describing the GATE simulations, with more limited discussion in the sections for the remaining 4 cases.

Before presenting the results, however, it is important to decide on a framework for conveying the independent aspects of (1) random uncertainty, which stems from reflectivity measurement error, and (2) systematic uncertainty, or bias. Under the assumption of small relative measurement error $\sigma_Z Z_{\text{obs}}^{-1} \ll 1$, which is ensured by our specification of the minimum detectable reflectivity, the errors in the observation space ($\boldsymbol{y}$) are Gaussian distributed. Furthermore, since the forward model is linear in the state vector, this Gaussianity is preserved in going from observation to state space ($\boldsymbol{x}$). Therefore, the estimators for the state vector and its covariance matrix in Eq. (14) describe a multivariate normal distribution from which a real observing system would sample. Note that because our inverse algorithm does not impose a prior as is done in the optimal estimation approach, the estimated state vector and covariance matrix are *not biased by the assumed prior distribution*. One approach often used in instrument simulator studies to assess the effect of random error on the inferred atmospheric state and its uncertainty is to inject random noise into the observations according to an error model like that in Eq. (6). However, this approach would be redundant in this case because of the detailed uncertainty quantification given by $\hat{S}_x$ and the Gaussianity properties described above. Furthermore, by not adding random noise to the observed reflectivity profiles, we can utilize the humidity retrieval mean values in $\hat{\boldsymbol{x}}$ to assess measurement bias on a grid-cell-by-grid-cell basis.





For these reasons, we proceed by simulating observations without the injection of random noise and subsequently analyzing the simulated retrievals by separately addressing the issues of systematic and random error. While the variability of humidity at a given height within the LES is small, the retrieval estimate of that quantity in the state vector $\hat{x}$ can vary widely from grid-cell to grid-cell due to changing hydrometeor fields. Therefore, by analyzing the variability of retrieved state mean values, we can assess the level of systematic error in the DAR measurement due to effects including non-uniform beam-filling, multiple scattering, incorrect humidity interpolation assumptions, and insufficiency of the linear-in-frequency fit term for mitigating the hydrometeor scattering biases. Then, we separately examine the retrieval random uncertainty and it's scaling with along-track averaging distance by combining sequential along-track grid-cells using a weighted mean and variance approach. Furthermore, to explore the systematic error that comes from assuming an exponential humidity interpolation function, we perform retrievals for each case using two different scale heights of $H_\rho = 1.5$ and 2.5 km.

Finally, it is important to recognize that while we can calculate from the LES a 3D snapshot of observed reflectivity values, the notional DAR system is not intended to be able to measure such a 3D field in a single along-track overpass. Instead, we treat the simulated 3D reflectivity field as providing a rich set of statistically independent vertical profiles that can be used to assess the accuracy and precision of measurements from a DAR that would sample similar clouds over an along-track distance that is typically far greater than the linear size of the LES domain. To this end, we will examine two different beam sampling strategies in the following analysis: One using a fixed, nadir pointing beam that executes sequential along-track overpasses in a raster-scan fashion across the LES domain; and a second approach where we assume that the radar possesses intelligent pointing capabilities and can arbitrarily target a single pixel in the cross-track ($x$) dimension for each along-track ($y$) position. For this latter approach, we envision that the spacecraft would be outfitted with an additional passive sensor, for instance a microwave imager, that can provide an estimate of cloud liquid water path in advance of the radar measurement, and can therefore feed-forward this information to the intelligently scanning antenna. While such a capability is highly idealized, it provides a useful benchmark for assessing the limits of an intelligently scanned system.

## 3.1 Deep convection: GATE

Forward simulated DAR observations for the GATE deep convection case are presented in Fig. 3. In general for the different LES cases, results are shown for the online frequency $f_2 = 174.8$ GHz because it experiences the most attenuation and therefore is the limiting factor for the 3-channel DAR system in terms of sampling. For instance, as shown in the vertical reflectivity profiles for all 3 frequencies in Fig. 3(d), the two offline channels $f_0$ and $f_1$ penetrate the entire cloud column down to the surface, while the online channel is fully extinguished at a height of about 2 km. The observed ocean surface NRCS in Fig. 3(a) reveals the horizontal structure of this deep convection scene, showing how regions near plumes contain significant hydrometeor burden that prohibits detection of surface echoes at $f_2$. Nevertheless, there is still substantial coverage in both the horizontal and vertical (see Fig. 3(c)) near these deep convective clouds. Finally, Fig. 3(d) provides a clear picture of the water vapor differential absorption signature in the observed multi-frequency reflecitvity profiles with continuous sampling at all frequencies from roughly 2 km to 12 km in height.





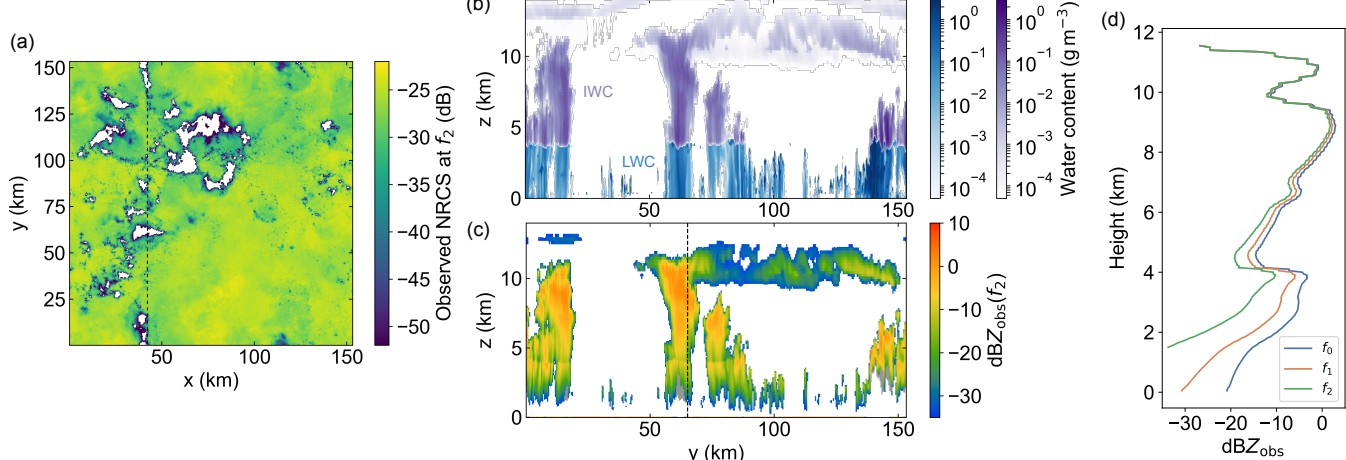

**Figure 3.** Selected results from GATE forward simulations. (a) Observed surface NRCS at $f_2 = 174.8$ GHz. Surface echos with SNR below the detection threshold appear as white, and generally correlate with convectively active regions with large LWC and IWC values. (b) 2D cross section of model condensed water profiles (ice and liquid) for the along-track path (dashed line) in (a). (c) Corresponding observed reflectivity profiles at 174.8 GHz. Gray regions in lower-tropospheric convective cores indicate regions where multiply scattered signal is at least 3 dB higher than the single-scattering reflectivity. We assume that these grid points can be filtered out before performing water vapor retrievals.

The results of the GATE retrievals are summarized in Fig. 4, where panels (a) and (b) pertain only to systematic measurement error, and panel (c) shows the scaling of random uncertainty, or precision, with along-track averaging for the two radar sampling approaches, fixed-nadir and intelligently scanned. We note that the GATE case is the only one for which multiple scattering features noticeably in the forward simulations, and we assume that range bins for which the multiply scattered signal is 3 dB higher than the single-scattering reflecitvity can be identified and filtered out before retrieval. The retrieval humidity values $x_2$ are classified according to 3 different types of DAR measurement columns: the "top column" between the radar and the first detected range bin; the "in-cloud" columns that include cloud returns at both the near and far ranges; and the "cloud to surface" column between the final cloud bin and the surface. In Fig. 4(a), the DAR retrieval (blue circles) corresponds to the average at each height of the in-cloud profiling retrievals that have a vertical resolution of 200 m. That is, partial column retrievals that are still in-cloud, but have a column lengths greater than 200 m are excluded as these do not represent high resolution, local humidity measurements. Importantly, the blue error bars in Fig. 4(a) *do not represent random measurement error*, but instead are the standard deviations of the mean retrieval values at each height for the entire LES domain. Therefore, these error bars provide a measure of the spread of in-cloud profiling biases, while the close agreement between the mean values and the conditionally sampled truth profile (open black circles) reveals that the mean bias is small.

The results of the IWV bias assessment for all 3 retrieval column types are shown in Fig. 4(b). Here the histograms correspond to retrievals performed using a humidity scale height of $H_\rho = 2.5$ km, with the median biases shown by the solid black lines. The black dashed lines show the median IWV biases that result from retrievals performed using the other scale height of



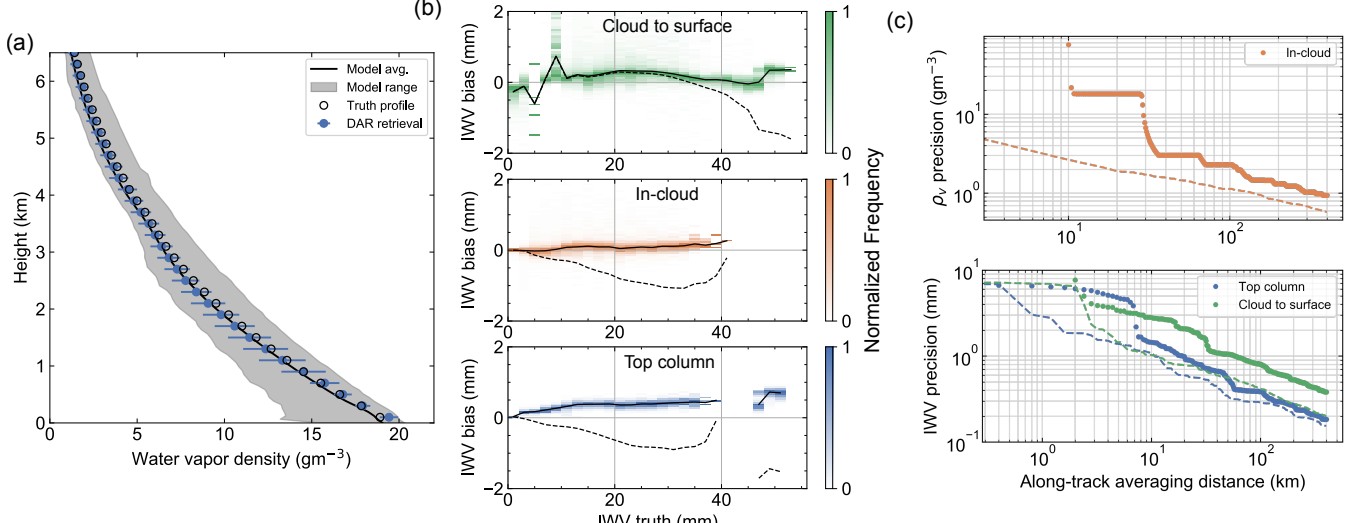

**Figure 4.** GATE DAR retrieval results for systematic (panels (a) and (b)) and random (panel (c)) error assessments. (a) Lower-tropospheric mean retrieved water vapor profile (blue solid circles) compared with model values, including the full domain mean profile (solid black line), range of model humidity at each height (gray shaded region), and the conditionally sampled truth profile (black circles), which is calculated using only grid cells for which the DAR retrieved a 200-m-resolution profile value. Note that the DAR retrieval error bars correspond to the standard deviation of mean retrieved values (i.e., elements of $\hat{x}$) at each height, and do not depict measurement uncertainty from random error. (b) 2D histograms of IWV bias from retrievals using $H_\rho = 2.5$ km as a function of truth IWV, for the three distinct DAR column retrieval types (top = lowest column between surface and last cloud return; middle = partial columns within/between cloud layers; bottom = uppermost column between top-of-atmosphere and first detected range bin). Each vertical histogram is normalized to have a peak value of 1 for display purposes. Solid black lines correspond to the median bias when using $H_\rho = 2.5$ km, while dashed lines are the same for $H_\rho = 1.5$ km. (c) Retrieval precision scaling with along track averaging, where the "in-cloud" curves (top panel) correspond to a profiling height of 3.9 km. Dashed lines correspond to an intelligently scanned radar (see text). Solid circles represent the humidity precision scaling for a hypothetical along-track path that pieces together nadir-pointing transects for $x = 21.1, 42.3, 63.1,$ and $84.3$ km.

$H_\rho = 1.5$ km. For this analysis, we include all columns of the in-cloud type, and do not restrict them to be of 200 m length. The magnitude of the median bias never exceeds 2 mm IWV for either value of $H_\rho$, and the results for $H_\rho = 2.5$ km show the exceptional accuracy of this DAR measurement method in capturing both in-cloud and clear-air water vapor columns, with biases rarely exceeding 0.5 mm.

We assess the scaling of in-cloud, 200-meter profiling precision with along-track averaging distance (Fig. 4(c)) by performing a weighted average of retrievals at a specific height of 3.9 km using the both fixed-nadir (orange circles) and intelligently scanned (orange dashed line) approaches. The result is that after 100 km of along-track averaging, these sampling approaches yield random humidity uncertainties of 2 gm$^{-3}$ and 1 gm$^{-3}$, respectively. It is important to note that the humidity profiling resolution and random uncertainty can be traded off in a linear fashion, meaning we would expect profiles with 400 meter resolution to exhibit half the uncertainty shown here. Because the cloud-to-surface and top columns are typically measured



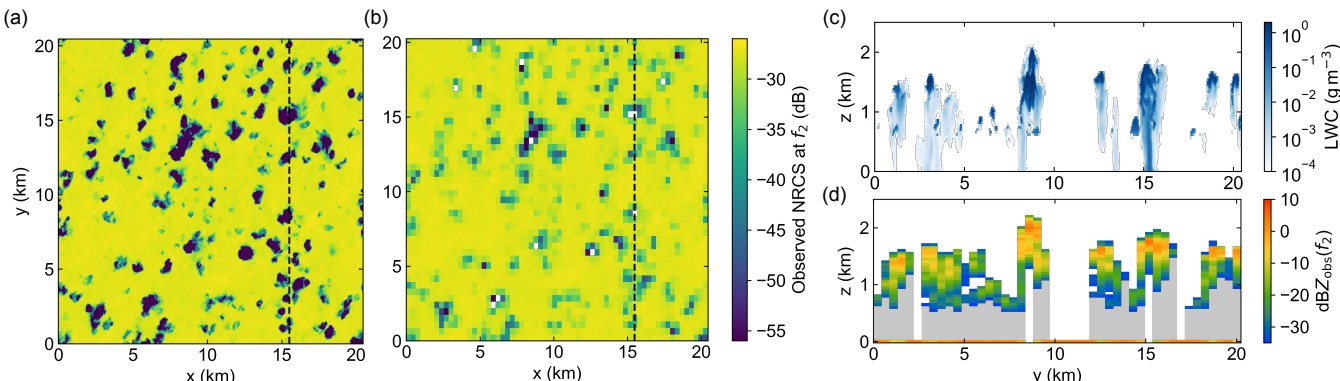

**Figure 5.** Selected results from RICO forward simulations. Simulated observed ocean surface NRCS at 174.8 GHz before (a) and after (b) filtering for instrument horizontal resolution and detection sensitivity. (c) LWC profiles for the along-track transect indicated by the dashed line in (a) and (b). (d) Corresponding observed reflectivity profiles at 174.8 GHz. Positions where cloud-to-surface partial column water vapor retrievals can be performed are highlighted with gray shading.

over a much larger differential absorption baseline, the amount of averaging necessary to reach a specified relative uncertainty is reduced relative to the in-cloud profiling. Here, retrieval precisions for both column types are at the 1-mm level after only 10 km of along-track averaging. These column IWV retrieval capabilities could provide unprecedented observations of horizontal water vapor variability in deep convective cold pools.

### 3.2 Precipitating shallow cumulus: RICO

Forward simulated DAR observations for the RICO case with precipitating shallow cumulus clouds are presented in Fig. 5. Because of the high horizontal resolution of the LES relative to the radar footprint, we compare the simulated observed ocean surface NRCS both before (panel (a)) and after (panel (b)) averaging over the instrument beam pattern. In this case, the reduced hydrometeor burden relative to the deep convection regions of the GATE simulation results in nearly ubiquitous sampling of the surface at $f_2$. In Fig. 5(c) and (d), we show the model cloud and rain LWC profiles and corresponding observed reflectivity for the fixed-nadir path shown by the dashed line in panels (a) and (b). Even at the highly absorbing online frequency $f_2$, the radar is able to sample almost a large fraction of the LWC field within the PBL. Furthermore, we highlight using grey shading in Fig. 5(d) the along-track positions for which the cloud-to-surface column will be retrieved.

The retrieval results for the RICO case are presented in Fig. 6 in a nearly identical format to the GATE simulations of the previous section. Because of the reduced sampling of IWV bias vs. truth relative to GATE, which is a result of the decreased cloud and precipitation amount, we present the bias results as scatter plots instead of histograms for both the $H_\rho = 1.5$ km (Fig. 6(b)) and $H_\rho = 2.5$ km (Fig. 6(c)) cases. An important feature of the in-cloud profiling retrievals to highlight is the ability of the DAR to resolve the inversion layer thickness and capture the water vapor profile near the top of the PBL, where there is a rapid change with height of the hydrometeor fields and therefore strong potential for systematic error. In fact, in separate end-to-end simulations run using a radar range resolution of 200 m (not shown), we find a systematic, large negative



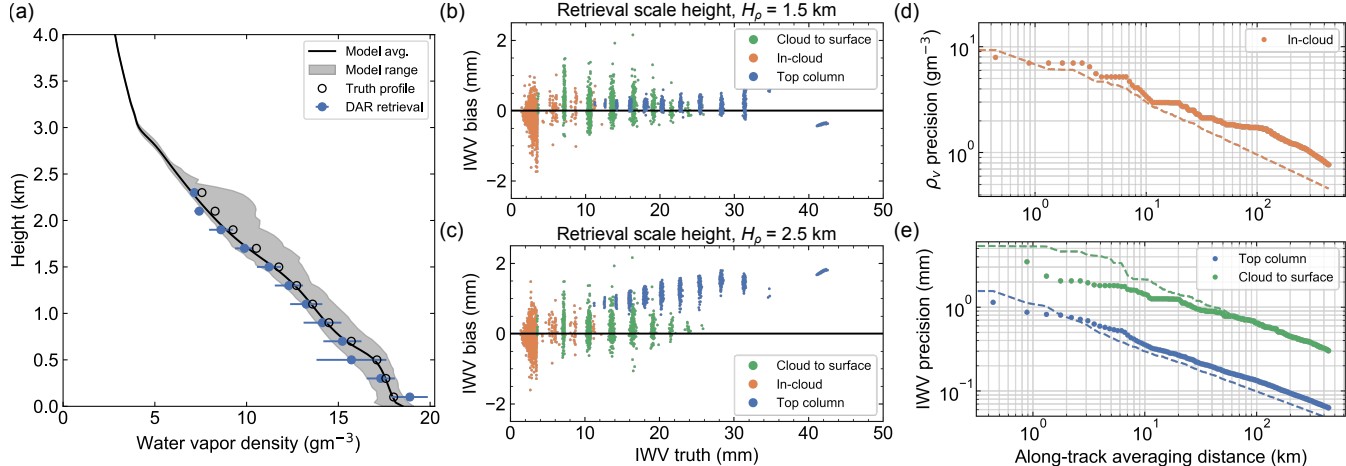

**Figure 6.** RICO DAR retrieval results. (a) Comparison of mean-retreived and model in-cloud humidity profiles, depicted as in the GATE case (Fig. 4(a)). (b-c) IWV retrieval bias vs. truth for the three distinct column types and two different humidity retrieval scale heights of 1.5 and 2.5 km. (d-e) Retrieval precision scaling with along-track averaging, similar to Fig. 4(c), except sequential along-track transects are executed by cycling through all possible cross-track ($x$) positions.

bias in this regime, which is due to non-uniform hydrometeor filling of range bins near the cloud top. Because the retrieval

10  algorithm interprets reflectivity signals as representing an average throughout the bin, this non-uniformity tends to diminish the differential absorption between the first two cloud bins, causing the negative humidity bias. However, by oversampling the humidity retrieval resolution by a factor of $O = 4$ here (see Section 2.4), this non-uniform range-bin filling issue is mitigated, and biases significantly reduced.

By comparing the retrieval biases for $H_\rho = 1.5$ and 2.5 km in Fig. 6(b) and (c), we see that the humidity scale height choice

15  mostly affects the top column retrievals. This is because the in-cloud and cloud-to-surface columns are of such short length that the choice of humidity interpolation function is unimportant, and the retrieval is only sensitive to bulk IWV in the partial columns. Furthermore, even if one uses an inappropriate retrieval scale height of 2.5 km, the top column biases remain fairly small, with a clear sky total column water vapor (TCWV) bias of at most 2 mm, or about 5% (see the rightmost cluster of top column retrievals in Fig. 6(c)).

20  The retrieval precision analysis in Fig. 6(d) and (e) reveals similar performance as in GATE, where in this case we present in-cloud precision results at a profiling height of 1.3 km. However, because in RICO the clouds are limited to the lowest 2.5 km of the atmosphere, the top column retrievals are always associated with a column that is nearly the depth of the atmosphere, resulting in reduced uncertainty relative to GATE. In this case, it is only necessary to average along-track for 1 km to achieve a top column precision of 1 mm IWV, and for 20-30 km to realize the same precision for the cloud-to-surface column, depending

5  on the scanning strategy.





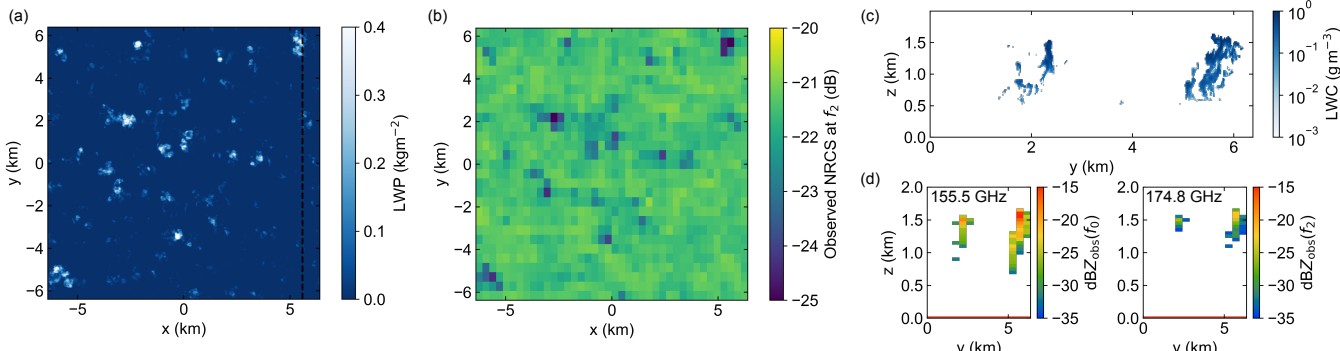

**Figure 7.** Selected results from BOMEX forward simulations. (a) Liquid water path (LWP) map for shallow cumulus scene highlighting the low cloud area fraction. (b) Simulated observed ocean surface NRCS at 174.8 GHz at the radar resolution. (c) Cloud LWC profiles corresponding to dashed line in (a). (d) Observed reflectivity profiles at 155.5 (left) and 174.8 (right) GHz.

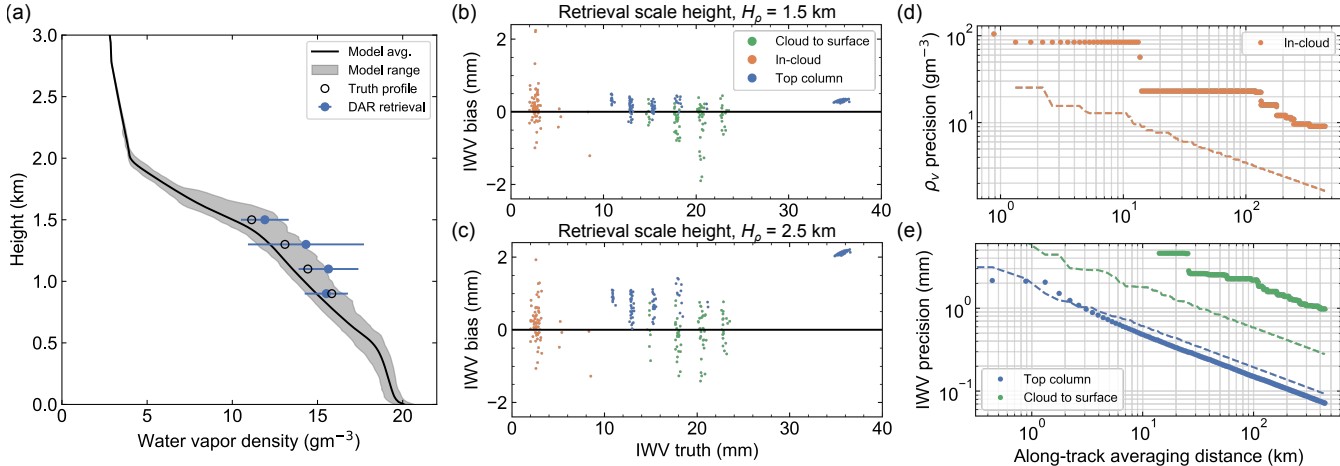

**Figure 8.** BOMEX DAR retrieval results. (a) Comparison of mean-retreived and model in-cloud humidity profiles, depicted as in the GATE case (Fig. 4(a)). (b-c) Retrieved IWV bias vs. truth for assumed humidity scale heights of 1.5 and 2.5 km. (d-e) Retrieval precision scaling with along-track averaging, where the in-cloud results correspond to a retrieval height of 1.5 km.

### 3.3 Non-precipitating shallow cumulus: BOMEX

The BOMEX case of non-precipitating shallow cumulus clouds is a particularly challenging one for in-cloud DAR observations, and for cloud radar in general, due to the relatively low volume cloud fraction and therefore limited sampling. This fact is highlighted by the forward simulation results in Fig. 7, where the 2D map of cloud liquid water path (LWP) in panel (a) suggests that the radar will primarily encounter clear-air columns down to the ocean surface, especially without intelligent scanning capabilities. In Fig. 8(c) and (d), we show the cloud LWC profiles and observed reflectivities at 155.5 and 174.8 GHz for the dashed line path in panel (a).





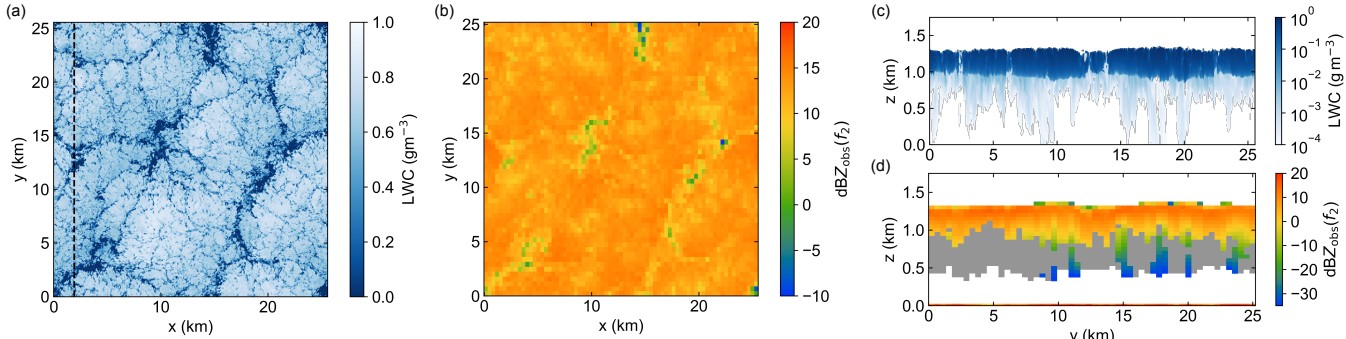

**Figure 9.** Selected results from VOCALS forward simulations. (a) Model LWC field for $z = 1250$ m. (b) Corresponding observed reflectivity map at 174.8 GHz for the range bin centered at 1250 m height. (c) Vertical LWC profiles for the along-track path indicated by the dashed line in (a), with drizzle clearly visible below cloud base, and (d) corresponding 174.8 GHz observed reflectivity profiles. The gray shaded region in (d) indicates where the multiple scattering mask has been applied, using the same criteria as in the GATE case.

The BOMEX retrieval results are summarized in Fig. 8 in a format identical to that for RICO. Unsurprisingly, the limited in-cloud sampling in this case results in increased variance of the profiling bias shown in panel (a). Furthermore, the in-cloud profiling precision is diminished, even for the intelligently scanned system, which reaches a precision of 2 gm$^{-3}$ only after an along-track averaging distance of 300 km. We conclude from these results that any hope of obtaining in cloud profiles in non-precipitating shallow cumulus requires an intelligent scanning capability. Furthermore, even a scanned measurement is unable to resolve the PBL inversion as can be done in the RICO case due to the weak cloud scattering from within the inversion layer. Nevertheless, the notional DAR can retrieve cloud-to-surface and top column IWV values with good accuracy and precision, especially for the intelligently scanned system, with biases below 0.5 mm for $H_\rho = 1.5$ km, and averaging distances of 30 and 3 km necessary to achieve 1 mm IWV precision, respectively.

### 3.4 Drizzling stratocumulus: VOCALS

The VOCALS case represents the ubiquitous light precipitation and drizzle that is found in marine stratocumulus, for example in areas of open cellular convection. Because of the increase in area cloud fraction relative to shallow cumulus, and general horizontal homogeneity of the cloud field, both VOCALS and DYCOMS-II simulations will reveal little difference between a fixed-nadir and intelligently scanned radar system. Furthermore, the position of the cloud layer atop a shallow PBL with strong temperature inversion and hydrolapse permits a natural division of the atmospheric column into separate regions for the DAR retrieval. Specifically, the top column between the radar and the stratocumulus cloud top always provides a bulk measure of the amount of water vapor in the dry, free troposphere, while the cloud-to-surface column of variable depth $\leq 1$ km quantifies the IWV in the moist PBL.

The forward simulation results for the VOCALS case are detailed in Fig. 9. In panel (a), we show the model LWC at the peak cloud height of 1.25 km, highlighting the high area cloud fraction in this case, with resulting observed reflectivity map at





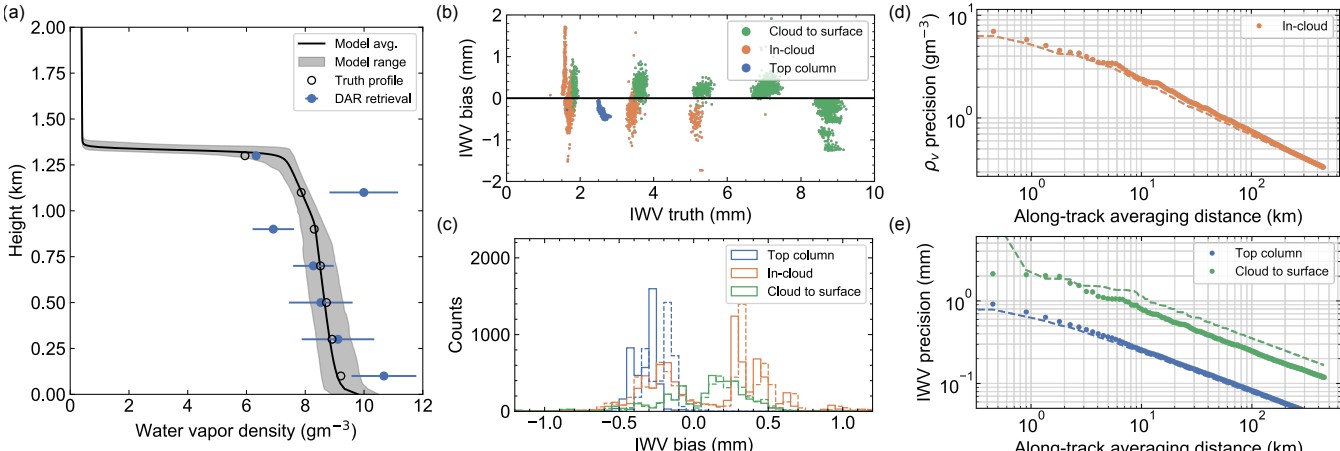

**Figure 10.** VOCALS DAR retrieval results. Panels (a), (b), (d), and (e) are as in Figs. 6 and 8, while (c) shows retrieval bias histograms for $H_\rho = 1.5$ km (solid lines) and $H_\rho = 2.5$ km (dashed lines). In (d) and (e), note the insensitivity of the precision to choice of scanning strategy (circles = fixed-nadir, dashed line = intelligently scanned) because of the horizontal uniformity of the cloud field.

the same height in panel (b). Even at the online frequency $f_2$, all horizontal columns contain a significant cloud detection. Fig. 9(c) and (d) show profiles of model cloud and drizzle LWC profiles and corresponding observed reflectivity for the along-track path indicated in panel (a) with a dashed line. Because of the reduced water vapor burden in the free troposphere relative to the previous cases of deep and shallow convection, there is greatly reduced atmospheric attenuation at $f_2$ permitting increased cloud sampling in the PBL. Note that as with the GATE case, we apply a multiple scattering mask to the simulated observations, where we assume that range bins with a multiply scattered signal that is 3 dB higher than the single scattering value can be filtered out. Regions where this mask has been applied are indicated with gray shading in Fig. 9(d), and are likely the result of multiple scattering from drizzle droplets within the cloud layer.

The VOCALS retrieval results are presented in Fig. 10. While in-cloud profiling is not expected to be a major strength of DAR for stratocumulus clouds, the retrievals reveal quite promising bias and precision capabilities in this regard, with random uncertainty at the 1 $gm^{-3}$ level after 50 km of along-track averaging (see Fig. 10(c)). Furthermore, the free tropospheric column and cloud-to-surface column results, with or without intelligent scanning, suggest that the notional DAR can provide very useful IWV estimates in this drizzling stratocumulus scene that could, for instance, be used to constrain near-surface humidity.

### 3.5 Non-drizzling stratocumulus: DYCOMS-II

The forward simulation results for the final case, DYCOMS-II, involving a non-drizzling stratocumulus cloud layer of thickness ≈ 200 m are shown in Fig. 11. As for the VOCALS case, we plot the model cloud LWC field in Fig. 11(a) for the peak cloud height of 820 m, and the corresponding radar reflectivity map in Fig. 11(b), revealing that the radar detects cloud in each sampled atmospheric column. From the observed reflectivity profiles in Fig. 11(d), we can see that each retrieval will feature

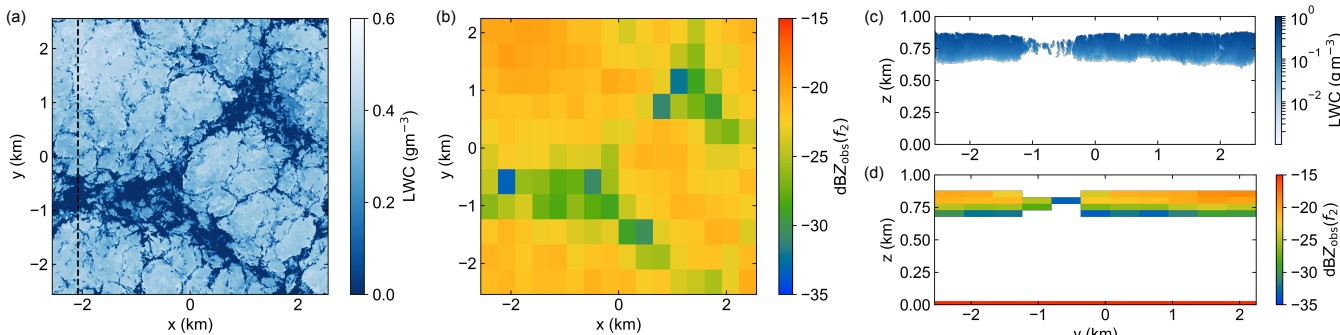

**Figure 11.** Selected results from DYCOMS-II forward simulations. (a) Model cloud LWC field for $z = 820$ m. (b) Corresponding observed reflectivity map at 174.8 GHz for the range bin centered at 800 m height. (c) Vertical LWC profiles for the along-track path indicated by the dashed line in (a) (note the reversed color scale with respect to (a)), and (d) corresponding 174.8 GHz observed reflectivity profiles.

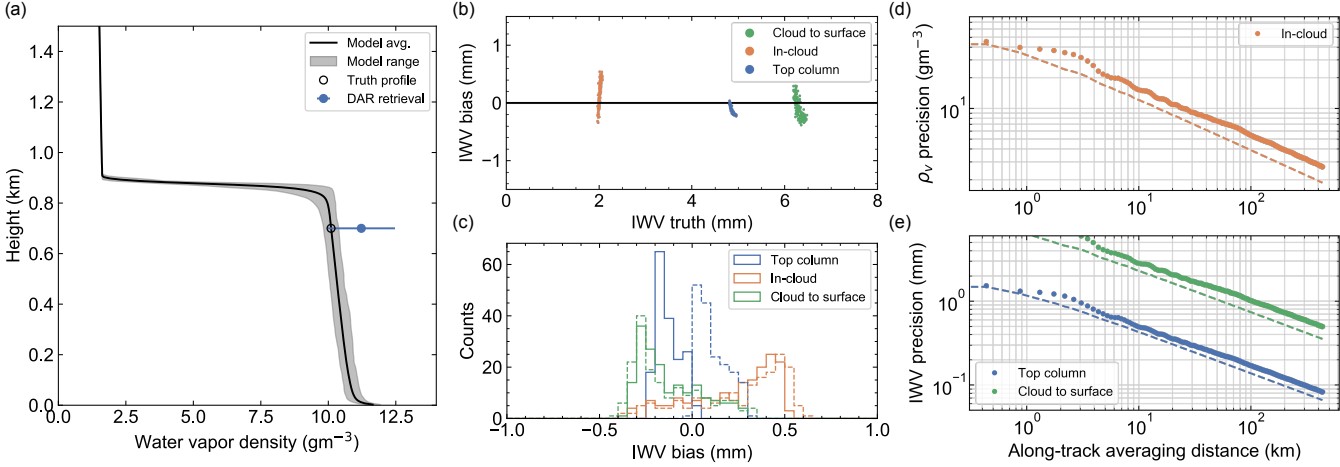

**Figure 12.** DYCOMS-II DAR retrieval results. Because of the limited vertical extent of cloud volume, every horizontal grid cell features just a single in-cloud profiling retrieved value, and identical upper- and lower-tropospheric column lengths. All panels have identical descriptions to those in Fig. 10.

distinct water vapor quantities: the free tropospheric column between the radar and cloud top; a single in-cloud humidity measurement of 200-m vertical resolution; and a cloud-to-surface column of length 600 meters representing the sub-cloud bulk humidity in the PBL.

The results of the retrieval are presented in Fig. 12 in an identical format to the VOCALS case, where we again see that the added utility of an intelligently scanned radar is minimal due to the high area cloud fraction. However, due to the reduced SNR of the cloud targets for the non-drizzling stratocumulus case, the in-cloud profiling and sub-cloud IWV precision feature a less favorable scaling with along-track distance. In realistic measurement scenarios the required $> 100$ km averaging distance may suffice due to the typically large horizontal extents of stratocumulus cloud structures.





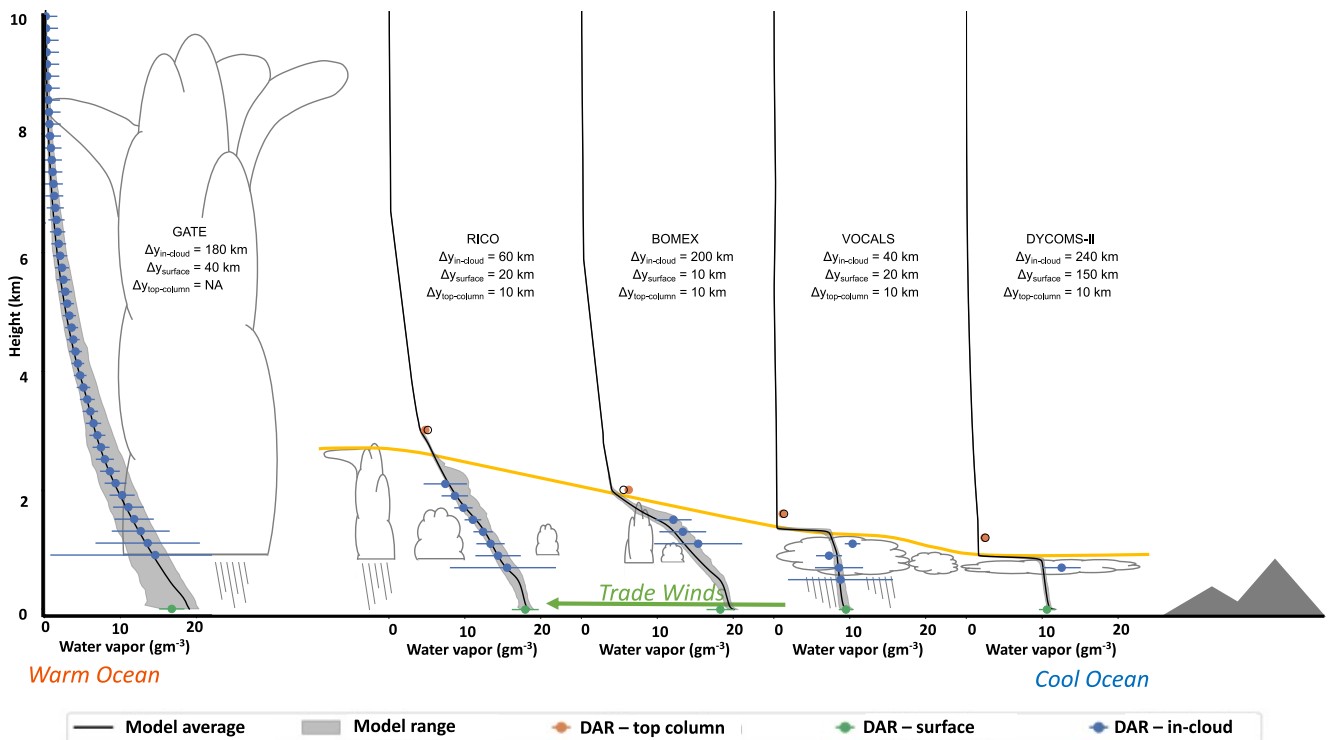

**Figure 13.** Summary of humidity retrieval capabilities for an intelligently scanned DAR system displayed within the context of the subtropical-to-tropical cloud transition regimes. For each LES case, we present the in-cloud profiles and surface humidity estimates calculated using the cloud-to-surface partial column with error bars representing $1\sigma$ random error. In-cloud estimates in the lower troposphere with relative uncertainty greater than 100% are excluded. The along-track averaging distances for the different retrieval quantity estimates are listed above. The top column representative water vapor density value (orange circles) is compared to the model truth value (open black circles).

## 4 Discussion

In the previous section, we presented the basic outputs of the end-to-end simulations and analyzed them from a statistical perspective with a focus on disentangling systematic and random retrieval uncertainty. Here, we provide example science case studies for utilizing the DAR observations, and begin the discussion with a summary of the DAR retrieval capabilities within the context of the subtropical to tropical cloud transition.

### 4.1 Retrieval performance summary

In order to synthesize a coherent picture of DAR retrieval capabilities for all 5 LES cases, we generate the following 3 quantities from the basic retrieval product of partial column IWV and its associated uncertainty, assuming an intelligently scanned radar system and variable along-track averaging distance. First, we transform the cloud-to-surface IWV retrieval for each along-track





position into an estimate of the surface humidity, assuming a well mixed boundary layer for all cases except GATE, where the partial columns often extend from the anvil cloud bottom to the surface, and therefore would not be well captured by a well-mixed assumption. Therefore, for the GATE case we assume an exponential humidity distribution with a scale height of 2.5 km. Second, we again filter the in-cloud retrievals for only those with 200 m vertical resolution. Third, we transform the top column IWV estimate into a representative estimate of the humidity 500 m above the highest in-cloud profiling position, for which we use the same exponential profile and humidity scale height as was used in the retrieval. Note that it should be expected that this third quantity does not agree with the model humidity value at the same height, because the upper tropospheric humidity profile shape is typically not well captured by an exponential. These three measured quantities and their uncertainties are then combined using a weighted mean and variance approach for sequential along-track grid cells to produce average observed profiles with associated random uncertainty.

Fig. 13 shows the results of this analysis, with an overlaid schematic depiction of the marine trade-wind subtropical to tropical cloud transition. The variable along track averaging distances, indicated by the respective $\Delta y$ values in the figure, are determined based on the following criteria. For the surface and upper tropospheric humidity estimates, the averaging distance that produces a 10% relative measurement error is used, with a minimum distance of 10 km imposed to ensure that potential systematic error is also accounted for. Then, for the in-cloud profiles, we again use the along-track distance that produces a specified relative uncertainty for at least one profiling height, but vary this number depending on the case to account for differences in cloud sampling and SNR. The minimum relative uncertainty criteria for the 5 cases in Fig. 13 are as follows: 15% (GATE), 10% (RICO), 20% (BOMEX), 10% (VOCALS), and 20% (DYCOMS-II).

These results constitute a concise summary of the extensive orbital simulations for the notional DAR instrument and what it's humidity measurement capabilities would be in these various cloud scenarios. It is important to note that both the high-resolution in-cloud profiles and the high-precision sub-cloud humidity observations in Fig. 13 are inaccessible with other sensor platforms, with the possible exception of GNSS radio occultation (RO) techniques. Specifically, GNSS RO systems can provide high resolution vertical profiles of refractivity in clouds, but must partition that refractivity signal between temperature and water vapor. Furthermore, GNSS RO retrievals struggle to profile below strong inversions due to the associated hydrolapse and in general measure an average over clear air and cloudy regions with 100-km-scale horizontal resoultion. DAR, on the other hand, is not affected by sharp thermodynamic profile gradients, and can utilize the surface echos to probe water vapor all the way to the surface.

## 4.2 Constraining near-surface RH

The surface relative humidity (RH) is a critical quantity in determining PBL structure. It sets the lifting condensation level and thus the cloud base and governs the magnitude of the surface moisture fluxes. As an example application of the DAR observations, we examine the ability to constrain the near-surface RH using the surface humidity estimates from Fig. 13. Recall that these estimates themselves are derived from the cloud-to-surface IWV column retrievals by assuming a well-mixed boundary layer. Therefore, two independent potential sources of bias in estimating near-surface RH are the retrieval IWV biases for the cloud-to-surface column and deviations of the atmospheric profile from the well-mixed state. The results are



| LES case | $\rho_{v,\mathrm{DAR}}$ (gm$^{-3}$) | $\sigma_{v,\mathrm{DAR}}$ (gm$^{-3}$) | DAR RH (%) | DAR $\sigma_{\mathrm{RH}}$ (%) | $\partial\mathrm{RH}/\partial T$ (%/K) | Model mean RH (%) | Model std. dev. RH (%) |
|---|---|---|---|---|---|---|---|
| DYCOMS-II | 10.6 | 1.1 | 69 | 7 | -4.2 | 76.0 | 1.6 |
| VOCALS | 9.5 | 1.0 | 64 | 7 | -3.9 | 66.3 | 2.4 |
| BOMEX | 18.3 | 1.8 | 71 | 7 | -4.1 | 78.1 | 1.2 |
| RICO | 18.0 | 1.8 | 74 | 7 | -4.3 | 76.0 | 1.4 |
| GATE | 16.7 | 1.7 | 76 | 8 | -4.5 | 87.0 | 2.9 |

**Table 4.** Estimating near-surface water vapor density ($\rho_{v,\mathrm{DAR}}$) and RH using the cloud-to-surface partial column DAR humidity measurement. Reported DAR RH values assume perfect knowledge of surface temperature, and sensitivity to this assumption is quantified by the term $\partial\mathrm{RH}/\partial T$. Note that for the GATE case, the partial column often extends well into the middle and upper troposphere, and therefore the inferred surface RH is highly dependent on the assumed sub-cloud humidity profile shape. For all other cases, a well-mixed PBL is assumed for the shallow sub-cloud column.

presented in Table 4, where we use the DAR-inferred surface humidity $\rho_{v,\mathrm{DAR}}$ and assume perfect knowledge of the surface temperature from the model to derive the DAR RH estimate. Then, the sensitivity of this RH estimate to the surface temperature assumption is quantified by computing $\partial\mathrm{RH}/\partial T$. While there is clearly RH sensitivity in the DAR cloud-to-surface column measurement, the relatively large uncertainty of $\sigma_{v,\mathrm{DAR}}/\rho_{v,\mathrm{DAR}} \approx 0.1$ and noticeable biases compared to the model values, even when assuming perfect knowledge of temperature, imply that the sub-cloud DAR moisture observations should be applied judiciously to the problem of constraining surface RH. An obvious approach to improve the constraint from the sub-cloud measurements on RH would be to relax the assumption of the well-mixed sub-cloud layer, however this is beyond the scope of this paper.

### 4.3 Inferring in-cloud temperature

The second example application of DAR retrievals explored here involves utilizing the high resolution profiles of water vapor inside of clouds and precipitation to estimate vertical temperature profiles by assuming that the volume is saturated with respect to liquid water. In the low latitudes it is observed that the spatial variability in temperature is significantly smaller than the variability in water vapor. Therefore a constrain on the in-cloud temperature profile in the subtropical marine boundary layer will provide a strong constraint on the domain mean. For this application, we focus on the retrievals from the RICO case, which involves only liquid clouds and provides roughly 2 km of continuous humidity profiling within the PBL. Generally, we expect an RH of 100% inside of cloud layers, but for precipitating volumes this can deviate significantly from saturation. Of course, from the perspective of the DAR, it is difficult to unambiguously distinguish between radar returns coming from range bins that are mostly cloud versus mostly rain. Furthermore, within a single radar footprint, it is likely that both cloudy and rainy regions are averaged together at a single height, in which case the rain signal will likely dominate due to the much larger average particle size. Therefore, by using the average DAR humidity profile acquired after a large along-track averaging



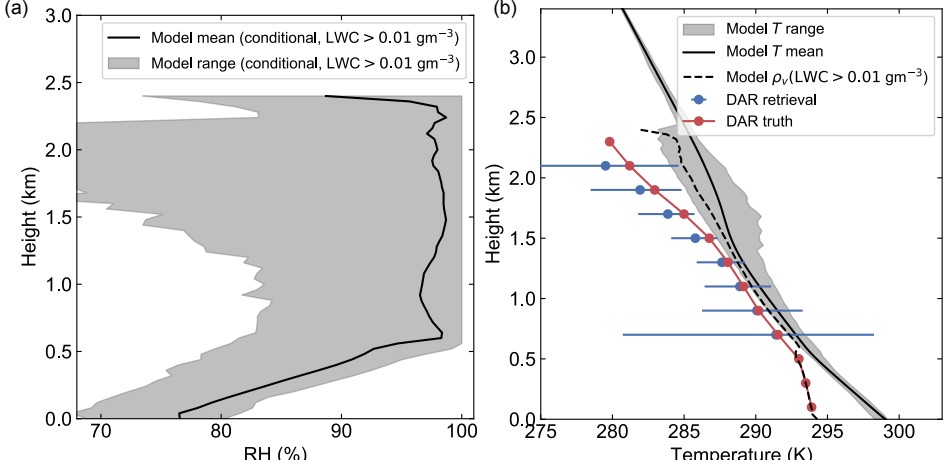

**Figure 14.** Using DAR humidity observations to estimate temperature profiles in RICO. (a) LES relative humidity profiles conditioned on LWP > 0.3 $\mathrm{kg\,m^{-2}}$, including the conditional mean (black line) and full model conditional range (gray region). (b) Comparison of model temperature profiles, including the conditional (LWP > 0.3 $\mathrm{kg\,m^{-2}}$) mean and full range, with the values inferred from DAR in-cloud humidity profiles with two different saturation assumptions of RH = 100 (blue circles) and RH = 90 (orange circles). Retrieved DAR humidity profile is identical to that in Fig. 13. Note that error bars for the RH = 90 DAR profile would be nearly identical to those for RH = 100, and are therefore omitted.

distance, we are likely to be combining fully saturated and sub-saturated volumes, and therefore potentially complicating the conversion from water vapor density to temperature.

In the fully saturated case, the temperature $T$ inferred from the DAR measurement at each height is simply the one that satisfies the relationship $\rho_v(z) = \rho_s(T)$, where $\rho_v(z)$ is the retrieved humidity value and $\rho_s(T) = e_s(T)R_v^{-1}T^{-1}$ is the saturated absolute humidity. Here $e_s(T)$ is the saturation vapor pressure curve for liquid water determined by the Clausius-Clapeyron relation and $R_v = 461.5\ \mathrm{K^{-1}kg^{-1}}$ is the water vapor gas constant. The results of this analysis using the RICO LES and simulated DAR retrievals are presented in Fig. 14(b), where we perform the conversion of humidity to temperature for 3 different profiles: the DAR retrieval, including random uncertainty, from Fig. 13; the conditionally sampled DAR truth profile from Fig. 6(a); and the model humidity profile conditioned on grid cells with LWC > 0.01 $\mathrm{gm^{-3}}$. Note that the temperature profiles inferred from the DAR retrieval and truth humidity profiles incorporate non-uniform beam filling effects, both in the horizontal and range dimensions, while the conditional model profile is at the model spatial resolution. Furthermore, the DAR truth and conditional model temperature profiles represent two different "ideal" measurements, with the former corresponding to a perfectly unbiased measurement made using the notional radar parameters in Table 3, and the latter a hypothetical measurement at the LES resolution requiring range and horizontal resolutions that are practically achievable.

For comparison with the humidity-inferred temperature profiles, we present in Fig.14(a) the LES mean RH profile, as well as the range at each height, both conditioned on LWC > 0.01 $\mathrm{gm^{-3}}$. Below the lifting condensation level around 500 m, we don't expect even the ideal humidity-inferred temperature measurements to agree with the model profile, as the liquid within





this region is purely precipitation falling through a sub-saturated volume. Furthermore, while the mean conditional model RH profile remains near 100% between the lifting condensation level (0.5 km) and cloud top (2.5 km), the conditional RH range is quite wide in this region, with the minimum conditional RH decreasing rapidly with height. We infer from this that there

exist in the LES grid cells that are horizontally displaced from the saturated updraft region, especially near the cloud top, that consist of rain, but not cloud, liquid water. This is likely the result of vertical wind shear within the model forcing a horizontal separation between updrafts and downdrafts, and has important consequences for the DAR measurement. Specifically, because the plumes are not highly resolved by the radar footprint, there will be many instances in which a single footprint encompasses updraft and downdraft regions, and therefore both sub-saturated and fully saturated volumes. However, because the radar

backscatter from precipitation particles is so much larger than for cloud droplets, the measured signal is heavily weighted towards the sub-saturated region within the footprint. This non-uniform beam filling bias towards lower humidity implies a negative temperature bias when assuming that the local environment is fully saturated, leading to the divergence of the DAR retrieval and truth profiles from the model values above 1 km in Fig. 14(b).

While it is theoretically possible to mitigate this negative temperature bias by decreasing the radar footprint size significantly,

such a solution would require an unrealistically large antenna. Therefore, for practical applications, it would be necessary to incorporate knowledge of the degree of non-uniform beam-filling and precipitation amount in order to adjust the RH = 100% assumption as a function of height within the cloud layer. Furthermore, this approach in general requires knowledge of the lifting condensation level — which could in principle be estimated from the DAR cloud-to-surface column IWV — in order to identify the sub-saturated region below the cloud base. Note, however, that we do not expect that similar non-uniformity biases

will occur in non-precipitating clouds because all of the radar reflection will be coming from saturated volumes. Therefore, much care must be taken in inferring in-cloud temperature from DAR observations to account for the presence of precipitation. In fact, such studies would benefit greatly from airborne DAR observations of real precipitating liquid clouds with coincident, in-situ observations of thermodynamic profiles from, for instance, dropsondes. In this case, the very high horizontal resolution offered by the airborne system would allow for systematic study of this non-uniform beam filling effect as a function of

horizontal averaging by combining along-track observations to form a larger effective footprint of variable size.

## 5  Conclusions

Spaceborne DAR holds considerable promise to fill gaps in the existing observing system for water vapor measurements, especially in the presence of clouds and precipitation. Through extensive LES and end-to-end radiative modeling, we have provided a high-fidelity assessment of the spatial coverage and systematic and random uncertainties that can be expected from a G-band DAR with realistically attainable performance metrics. Basic scaling arguments based on existing airborne G-band radar technology, as well as output from the orbital simulator suggest that a radar transmit power of at least 200 W and antenna diameter of 2 m are necessary for achieving adequate sampling of all cloud types explored in this work. Despite a fine range

resolution of 50 m, which is necessary for mitigating non-uniform range-bin filling biases from the water vapor retrievals,



a radar with these parameters is able to probe well into deep convection, as well as sensitively detect non-drizzling marine stratocumulus cloud layers with a thickness of only 200 m.

The notional DAR frequency channels are most sensitive to PBL humidity, with high-value water vapor retrieval products including in-cloud vertical profiles at 200 m resolution, sub-cloud IWV utilizing the strong surface reflection, and an upper
tropospheric IWV column between the radar and first cloud target. Additionally, the system provides a high precision measurement of total column water vapor when clouds are absent, which can be performed over land or ocean. As a general summary of the DAR capabilities for the 5 different LES cases, we find that in-cloud profiles with 10-20% relative uncertainty can be retrieved for horizontal resolutions of 100-200 km, depending on the cloud type, and that upper tropospheric and sub-cloud IWV measurements with 10% uncertainty require along-track averaging on the scale of tens of km. For all retrieval products,
we find the systematic uncertainty to be generally much smaller than the random error, thanks to the improved spectral fitting routine that incorporates a third DAR frequency and new fit parameter (Battaglia and Kollias, 2019).

In addition to a detailed retrieval performance assessment, we have outlined several potential science applications of DAR observations, including the estimation of near-surface relative humidity using the sub-cloud IWV column, and inference of in-cloud temperature profiles using the DAR humidity profiles and assumption about the average saturation state. While much
work remains to explore the limits of DAR for studies like these, the results demonstrate useful signals in the DAR observations for these applications. Beyond these examples and the basic utility of high-vertical-resolution water vapor information for modeling and assimilation applications, there are exciting opportunities to exploit DAR observations for probing the horizontal variability of water vapor in deep convective cold pools below the cloud anvil. Both continued simulation studies and instrument deployment on high-altitude airborne campaigns with in-situ validation are essential for understanding the applicability of DAR
observations to these science questions.

*Code availability.* The radar simulator code used in this work will be made available upon request.

*Data availability.* The large eddy simulation (LES) output files will be archived at the JPL Climate Center website, pending an interval for external release.

**Appendix A: Pulse-to-pulse correlations**

In this appendix, we recount the derivation of the Doppler frequency, or equivalently velocity, spectrum and associated pulse-to-pulse correlations that result from the combined effects of the finite antenna beam width and fast satellite motion. Our
5  approach is similar to previous treatments (e.g., Tanelli et al. (2002)) but with modified notation to elucidate only the essential points. To begin, we examine the power spectral density observed by the spaceborne receiver due to radar backscatter from a





stationary, volumetric target at a specific range $r$, assuming a transmit frequency of $f_0$:

$$S(r,f) \propto \int\limits_{-\infty}^{\infty} \int\limits_{-\infty}^{\infty} \int\limits_{r-\Delta r/2}^{r+\Delta r/2} dr' dy' dx' \eta_{\text{obs}}(f_0, \boldsymbol{r}') |F(f_0, x', y', r')|^4 \delta(f - f_0 - \delta f_D(y', r')). \tag{A1}$$

Here we have again used the relation $z' \approx R - r'$ due to the narrow width of the beam, and have assumed that the range
weighting function can be approximated by a box of width $\Delta r$. Note that satellite-motion-induced Doppler shifts are introduced
through the integration over the along-track distance variable $y'$ using a delta function in frequency, where the shift itself is
given by:

$$\delta f_D(y', r') = \frac{2v_{\text{sat}}}{\lambda_0} \sin \theta(y', r')$$
$$\approx \frac{2v_{\text{sat}} y'}{\lambda r'}. \tag{A2}$$

Next, we assume a Gaussian beam shape with one-way intensity pattern

$$|F(f_0, x', y', r')|^2 = \exp\left(-\frac{x'^2 + y'^2}{r'^2 \theta_0^2(f_0)}\right), \tag{A3}$$

where $\theta_0 = C_\theta \lambda_0 / D_a$ and $C_\theta$ is a constant of order 1 that depends on the specific antenna implementation. Here, we assume a
beam with $C_\theta = 0.41$. Then, the power spectral density in Eq. (A1) becomes, up to a constant multiplicative factor,

$$S(r,f) = S(r, f_0) \exp\left(-\frac{(f - f_0)^2}{2\sigma_f^2}\right), \tag{A4}$$

where the spectrum width is given by

$$\sigma_f = \frac{v_{\text{sat}} \theta_0}{\lambda_0} = \frac{v_{\text{sat}} C_\theta}{D_a}. \tag{A5}$$

Eq. (A5) reveals the important fact that the spectral width resulting from differential Doppler shifts in the along-track dimension
for a fixed antenna size $D_a$ is independent of the radar frequency. To get the corresponding width of the Doppler velocity
spectrum, we use the relationships $\delta f_D = 2v/\lambda_0$ and $\sigma_v^2 = (\partial v / \partial \delta f_D)^2 \sigma_f^2$ to get $\sigma_v = v_{\text{sat}} \theta_0 / 2$.

With the form of the power spectral density in hand, we calculate the signal autocorrelation function by invoking the Wiener-
Khinchin theorem:

$$R(t) = \int\limits_{-\infty}^{\infty} S(r,f) e^{2\pi i f t} df$$
$$= R(0) e^{-2\pi(\sigma_f^2 t^2 - i f_0 t)}, \tag{A6}$$

where $t$ is the time lag between two sample sequences, and the power spectral density and autocorrelation functions in Eq.
(A6) are intended to represent statistical ensemble averages. Finally, recognizing the *signal voltage* autocorrelation coefficient



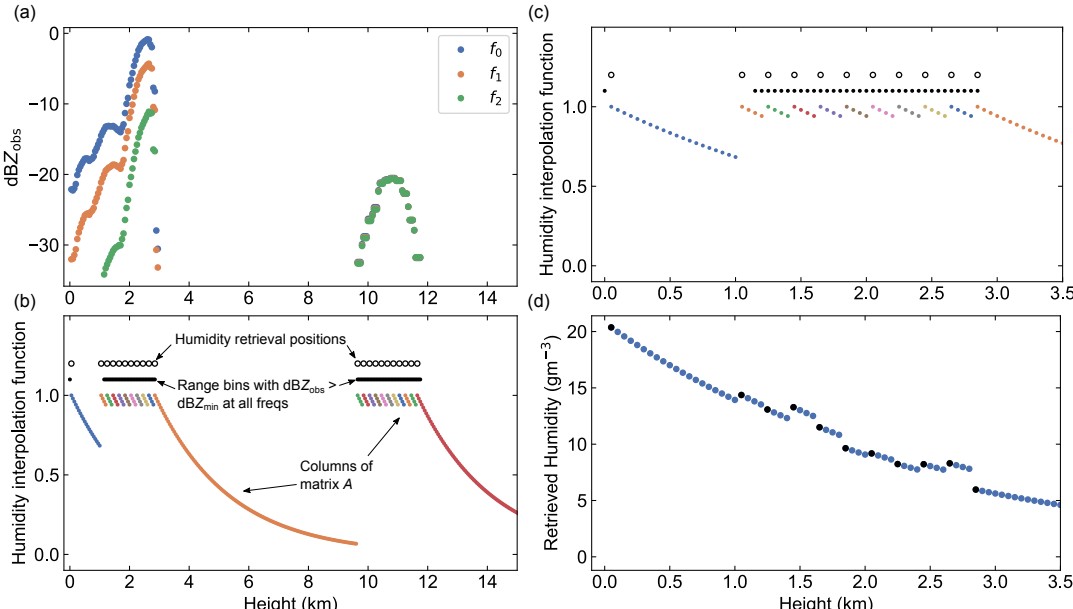

**Figure B1.** Humidity interpolation procedure for the DAR retrieval, using an example from the GATE case. (a) Vertical profiles of observed reflecitvity at the 3 DAR frequencies for the $y = 100$ km position in Fig. 3(c). In this case, the surface NRCS (not shown) is above the sensitivity threshold for all 3 frequencies. (b) Graphical depiction of the procedure for determining the projected measurement axis ($\boldsymbol{r} = P_r \boldsymbol{r}'$ - solid black circles) and humidity retrieval axis ($\boldsymbol{z} = P_z \boldsymbol{z}'$ - open black circles), as well as the humidity interpolation functions, which are the columns of the matrix $A$ (differently colored, partial exponential profiles). (c) Zoom-in of panel (b) for clarity. (d) Resulting retrieved humidity profile (uncertainty not shown), including the $N$ retrieved humidity elements of the state vector ($\boldsymbol{x}_2$ - black circles) and the corresponding interpolated humidity profile $A\boldsymbol{x}_2$ (blue circles).

5    as $\rho_s(t) = R(t)/R(0)$, we calculate the *power* autocorrelation coefficient according to $\rho_p(t) = |\rho_s(t)|^2 = \exp(-4\pi\sigma_f^2 t^2)$, and obtain an estimate of the number of statistically independent radar samples $N_i$ from (Papoulis, 1965; Doviak and Zrnić, 1993)

$$N_i = \sum_{m=-(N_p-1)}^{N_p-1} \frac{N_p - |m|}{N_p^2} \rho_p(mT_p), \tag{A7}$$

where $N_p$ is the number of pulses transmitted and $T_p$ is the pulse repetition interval. Note that the power autocorrelation coefficient provides a natural time scale for signal decorrelation, referred to as the time to independence, $\tau_i = (2\sqrt{\pi}\sigma_f)^{-1}$.

10   Then, Eq. (5) follows directly from Eq. (A7) and the definition of $\rho_p(t)$.

## Appendix B:  Retrieval humidity profile interpolation example

In this appendix, we utilize a single-footprint measurement from the GATE case to clarify details of the interpolation-based humidity retrieval algorithm detailed in Section 2.4. These ideas are explored in Fig. B1. The essential task of the grid-selection and interpolation procedure is to take as an input profiles of observed reflectivity at the 3 DAR frequencies, and calculate



in an algorithmic fashion the resulting measurement axis $\boldsymbol{r} = P_r\boldsymbol{r'}$ and humidity retrieval axis $\boldsymbol{z} = P_z\boldsymbol{z'}$. To determine the

5 measurement axis, we identify range bin locations for which the observed reflecitvity at all 3 frequencies is above the instrument detection limit $\mathrm{dB}Z_{\min}(f)$ (solid black circles in Fig. B1(b) and (c)). Then, we must decide on the humidity profile vertical spacing, which we force to be an integer multiple of the radar range resolution. Recall that in Section 2.4 we defined this integer as the range bin oversampling factor $O = 4$. Given the set of possible humidity retrieval positions $\boldsymbol{z'}$, we identify actual positions for humidity retrieval (open black circles in Fig. B1(b) and (c)) as those for which there is at least one element of $\boldsymbol{r}$ in the list $[z' - \Delta r, z', z' + \Delta r, z' + 2\Delta r]$. Note the offset of size $\Delta r$ of the first humidity position in Fig. B1(c) from the surface position. This is due to the fact that the forward model must compute absorption for a given level using information from levels above it, and therefore the lowest level in the humidity profile is one range bin displaced above the actual surface.

*Author contributions.*  R. Roy developed all elements of the radar simulator, developed all retrieval algorithms, and performed all of the data analysis. M. Lebsock conceived of the project and provided extensive guidance. M. Kurowksi conducted the LES studies and provided the LES output.

*Competing interests.*  The authors declare that they have no conflict of interest.

*Acknowledgements.*  This research was carried out at the Jet Propulsion Laboratory, California Institute of Technology, Pasadena, CA, USA,
under contract with the National Aeronautics and Space Administration.



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
