# Peer review of "Spaceborne differential absorption radar water vapor retrieval capabilities in tropical and subtropical boundary layer cloud regimes"

_Atmospheric Measurement Techniques, 2021_

## Referee Comment (RC1)

**Review for *Spaceborne differential absorption radar water vapor retrieval capabilities in tropical and subtropical boundary layer cloud regimes* by *R. Roy et al***

The study assesses the capabilities of spaceborne Differential Absorption Radar (DAR) for tropospheric humidity profiling in typical planetary boundary layer (PBL) cloud conditions along the transition from sub-tropical to tropical regimes. Five different LES model cases are used to simulate spaceborne radar measurements at 155.5, 168 and 174.8 GHz. DAR capabilities are evaluated based on the simulated measurements for each scene regarding the retrieval of in-cloud water vapor profiles, sub- and above-cloud water vapor amount, as well as IWV in clear-air scenes. Application examples are given for deriving near-surface RH and in-cloud temperature profiles. The authors analyze expected uncertainties and instrument requirements and conclude that spaceborne DAR holds the potential of filling current observational gaps of PBL profiling.

The study presents a suitable and novel framework to address the current observational gap of PBL profiling. It is an important contribution to the field with exciting results, and particularly timely as DAR and G-band radar technique are advancing. The study is clearly written and well presented. It comprises a very detailed presentation of the applied forward simulator and retrieval algorithm method, as well as an extensive results section. The amount of material makes the manuscript quite long. I have some major and minor comments of what is otherwise a convincing manuscript.

***General comments:***

1. DAR aims at filling observational gaps in PBL profiling. The authors should emphasize which type of variability can be resolved by their approach, and if this meets the needed requirements to characterize the analyzed regimes.
2. The use of a scale height for exponential humidity interpolation requires more justification. In trade wind conditions, the humidity profile strongly deviates from an exponential profile. The limitations should be further discussed and the impact of this assumption on the retrieval results should be quantified in more detail.
3. Intelligent scanning technique is a powerful tool for future satellite-based applications. When applying this sampling strategy, do the authors account for the tilted, off-nadir inclination angle and the resulting change of surface NRCS in their forward simulations? If not, a quantification of the impact should be added.
4. The authors add a third frequency to the standard 2-frequency DAR approach. More information should be provided regarding the choice of the channel. The authors should also highlight in which conditions and at which heights signal saturation occurs in one of the three channels, e.g. as function of hydrometeor and gas loading. Which conditions would be particularly favorable and which are most challenging for spaceborne DAR to measure in and fill existing observational gaps?
5. The manuscript contains a lot of material, at times distracting the reader from the main messages. For example, the authors develop and present a radar forward simulator in detail as tool for their analysis. Why did they not consider an established tool such as CR-Sim (Oue et al., 2020; https://gmd.copernicus.org/articles/13/1975/2020/) or PAMTRA (Mech et al., 2020; https://doi.org/10.5194/gmd-13-4229-2020/)? Changes should be highlighted, also regarding the simulator presented in Millán et al., 2020 (https://doi.org/10.5194/amt-13-5193-2020). Many details on radar simulation are extensively described by literature. Thus, the authors should
   (i)      move technical details in Sec. 2.2. to the appendix;
   (ii)      reduce their selection of Figures 5-12; e.g. by compressing the presentation of the simulated measurements and/or moving them to Sec. 2.2.3 or the appendix

*Specific comments:*

1. Two relevant contributions to this field are missing in the introduction: Lamer et al (2021), doi.org/10.5194/amt-14-3615-2021 ; Schnitt et al (2020) , doi.org/10.1175/JTECH-D-19-0110.1

2. p.3, ll. 16 More details should be provided on selection of 155.5GHz channel as third frequency. Why was the channel at 167 GHz (Roy et al, 2020) moved to 168 GHz here?

3. p.4, tab. 1: the following LES parameters should be added: vertical top height of LES simulation; longitude and latitude of domain center; considered time step; IWV variability within the domain. I suggest to indicate mean and standard deviation. IWV (or TCWV?) should be used consistently throughout the manuscript.  I also suggest to quantify the variability of $\rho_v$ and IWV in the respective simulations to give the reader a feeling on the expected variations of the respective moisture fields.

4. p.4, ll.5: A clarifying sentence should be added on how the different horizontal and vertical resolutions of LES and MERRA-2 were accounted for when completing the LES columns. A reference and specifications to the used MERRA-2 dataset are missing.

5. p.4, Sec. 2.2: also see major comment 5. The authors should highlight the advantages of their radar simulator compared to existing radar forward models.

6. p.8, l.17: Which 'relevant variables' were chosen?

7. p.9, tab.3: range indication for $dBZ_{min}$ and explanation for * in horizontal footprint line are missing.

8. p.11, Fig.2: I suggest to add the reflectivity profiles of $f_0$ and $f_2$ to the right panels of (a) and (b), respectively. Considering a detection threshold of -34dBZ as given in Tab 3, a detection of the profile as shown in (b) below 2km does not seem possible. A clarifying statement should be added on how saturated profiles are dealt with in the retrieval, and how often saturation occurs in the 5 different LES cases. How many profiles were simulated in total? CFAD diagrams could give the reader an additional overview on the simulated radar measurements.

9. p.15, ll.11: the authors should clarify how exactly they chose the profiles that were sampled when applying intelligent scanning sampling strategy.

10. p.15, ll.11: Do the authors think that an intelligent scanning technique could be, in a realistic application, combined with adjusting one of the frequencies to overcome sensor saturation for the expected conditions?

11. p.16, Fig.3: figure caption for panel (d) is missing.

12. p.17, ll.2; Fig. 4(c) (and 6,8,10d): The authors should add a clarifying comment how the $\rho_v$ precision changes at different heights (e.g. lower BL below 3.9km in the GATE case), or why the specific shown height was chosen. I suggest to add a clarifying statement, or to add the precision at different heights to the respective panel.

13. p.24, Fig.13: This figure summarizes the results of the intelligently scanning method nicely. How do these results differ when applying the analysis to the nadir sampling strategy? I suggest to address this in the text to uphold figure clarity.  The authors should add to their discussion whether the derived along track averaging distance is sufficient to resolve the expected horizontal water vapor variability.

14. p.25, ll.18: I am wondering if the required fulfillment of the 200m vertical resolution systematically excludes certain conditions – can the authors provide a threshold or typical cases in which this requirement is not fulfilled?

15. p.26, tab. 4: A column giving the surface temperature should be added.

16. p.26 ll.20: A reference for this statement should be added.

17. p.27, l.15: the authors should provide more information about the applied LWC > 0.01gm$^{-3}$ threshold and the in-cloud temperature derivation for thinner clouds. How does this threshold impact the presented results?
18. p.27: Fig 14b) is referenced before 14a).
19. p.28, conclusions: based on their experience with the present analysis, I encourage the authors to add a concluding remark how their simulated system would perform in different conditions such as at higher latitudes.

---

## Author Comment (AC1)

**Review for *Spaceborne differential absorption radar water vapor retrieval capabilities in tropical and subtropical boundary layer cloud regimes* by *R. Roy et al***

The study assesses the capabilities of spaceborne Differential Absorption Radar (DAR) for tropospheric humidity profiling in typical planetary boundary layer (PBL) cloud conditions along the transition from sub-tropical to tropical regimes. Five different LES model cases are used to simulate spaceborne radar measurements at 155.5, 168 and 174.8 GHz. DAR capabilities are evaluated based on the simulated measurements for each scene regarding the retrieval of in-cloud water vapor profiles, sub- and above-cloud water vapor amount, as well as IWV in clear-air scenes. Application examples are given for deriving near-surface RH and in-cloud temperature profiles. The authors analyze expected uncertainties and instrument requirements and conclude that spaceborne DAR holds the potential of filling current observational gaps of PBL profiling.

The study presents a suitable and novel framework to address the current observational gap of PBL profiling. It is an important contribution to the field with exciting results, and particularly timely as DAR and G-band radar technique are advancing. The study is clearly written and well presented. It comprises a very detailed presentation of the applied forward simulator and retrieval algorithm method, as well as an extensive results section. The amount of material makes the manuscript quite long. I have some major and minor comments of what is otherwise a convincing manuscript.

***General comments:***

1. DAR aims at filling observational gaps in PBL profiling. The authors should emphasize which type of variability can be resolved by their approach, and if this meets the needed requirements to characterize the analyzed regimes.

   Thanks for pointing out that we neglected to place DAR within the context of a larger observing system. The NASA PBL Decadal Survey Incubation Study Team recently released a report (https://science.nasa.gov/earth-science/decadal-pbl) in which they envision DAR as only one piece of a comprehensive core observing system containing also a DIAL and advanced hyperspectral IR and microwave sounders. The objective of the DIAL/DAR is to observe the vertical variability while the passive sounders provide the horizontal variability. We have added some text to this effect.

   Added Text Page 2, Line 11 (after 200m): "The advantage of the active DIAL/DAR approaches is to constrain the vertical variability in water vapor desired by the decadal survey (NASEM, 2018). It is anticipated that these observations would exist within a larger observing system where the requisite horizontal variability is observed by passive infrared and microwave systems."

2. The use of a scale height for exponential humidity interpolation requires more justification. In trade wind conditions, the humidity profile strongly deviates from an exponential profile. The limitations should be further discussed and the impact of this assumption on the retrieval results should be quantified in more detail.

   Added text Page 11, Line 24 (after cross section). 'We note that the assumption of the exponential profile is not necessary. We use this assumption here to demonstrate how a radar-only retrieval might perform in the absence of any additional information. In practice, in an operational environment a more detailed profile shape would be taken from a weather analysis system or from coincident observations from passive sounders. In fact, we show below that the derived column water vapor measurements below are

significantly sensitive to the exponential scale height motivating the future use of ancillary data in the DAR retrievals'.

We note to the reviewers that the sensitivity to the assumed scale height is shown in figures 6 and 8 for the RICO and BOMEX cases. We agree that the assumption of an exponential profile is often wrong, however we argue that in the absence of any other information (i.e. a DAR-only retrieval) an exponential scale height is the optimum general-scenario choice.

3. Intelligent scanning technique is a powerful tool for future satellite-based applications. When applying this sampling strategy, do the authors account for the tilted, off-nadir inclination angle and the resulting change of surface NRCS in their forward simulations? If not, a quantification of the impact should be added.

The simulations do not account for the change in the NRCS, however these effects are negligible. We have a paper in review on the NRCS at G-band (see figure below). We envision that a 40-50 km cross-track sampling range would be sufficient to address most of the sampling challenges for DAR associated with shallow convection, and this regime is our motivation for pursuing this approach. From 400 km altitude this sampling capability can be accommodated with roughly +/-3 degrees of scanning. The variation in NRCS within this narrow range of incidence angle is well modeled by a geometric optics approach and is on the order of 1-2 dB. Variations due to wind speed will be larger than this. In any case, in nearly all scenarios the retrieval uncertainty will not be affected by these small changes in surface reflection because the precision is independent of signal magnitude in the regime SNR >> 1. In cases where the surface SNR is not much greater than 1 the signal is generally completely attenuated by precipitation and there is no visible surface reflection at all.

Added text Page 13, Line 32 (after position) "We envision a cross-track scanning capability on the order +/- 3 degrees which from 400 km altitude provides ~42 km range in cross track sampling location. This relatively limited scanning capability would provide significant increases to cloud sampling opportunities in shallow convective cloud regimes while also limiting the technical complexity of a wide-angle scan. We note that we do not model the angular dependence of the NRCS in this study, which has limited sensitivity over this modest angular range (Roy et al., 2021)."

[Figure]

Observed NRCS over ocean from the airborne observations with VIPR over a very calm ocean surface in the Santa Barbara channel. Smooth curves show theoretical calculations using a geometric optics approach for various wind speeds and a Cox-Munk surface model. Note that the curves are very flat near nadir even in this very calm case and the sensitivity to angle decreases as wind speed increases. Finally, typical wind speeds over ocean are ~ 7 m/s

4. The authors add a third frequency to the standard 2-frequency DAR approach. More information should be provided regarding the choice of the channel. The authors should also highlight in which conditions and at which heights signal saturation occurs in one of the three channels, e.g. as function of hydrometeor and gas loading. Which conditions would be particularly favorable and which are most challenging for spaceborne DAR to measure in and fill existing observational gaps?

The architecture is largely constrained by regulations rather than physics. 174.8 GHz is the closest frequency to the water line at which we are able to transmit because of regulations. The single best thing we could do to enhance performance would be to move that on-line frequency slightly closer to 183 - but we cannot. We point the reviewer to page 1, line 15, which states 'The 3 transmit frequencies located at 155.5, 168.0, and 174.8 GHz are carefully chosen to lie in bands that do not feature international transmission restrictions (National Telecommunications & Information Administration (NTIA), 2015)'. 155.5 GHz is chosen because it lies within a small piece of available spectrum between 151.5 and 155.5 GHz that is not designated as a 'passive' transmission band. We chose not to elaborate further on these details because they are far outside the scope of the typical radar scientist or atmospheric scientist.

Added Text Page 3 Line 22: "The middle DAR channel used here differs slightly from the 167 GHz lower frequency channel used in past two-frequency DAR measurements with VIPR (Roy et al., 2020). This new location is chosen to minimize covariance between fit parameters in the three-frequency retrieval (see $a\_2$ and $a\_3$ in Eq. (6) and (7)).

5. The manuscript contains a lot of material, at times distracting the reader from the main messages. For example, the authors develop and present a radar forward simulator in detail as tool for their analysis. Why did they not consider an established tool such as CR-Sim (Oue et al., 2020; https://gmd.copernicus.org/articles/13/1975/2020/) or PAMTRA (Mech et al., 2020; https://doi.org/10.5194/gmd-13-4229-2020/)? Changes should be highlighted, also regarding the simulator presented in Millán et al., 2020 (https://doi.org/10.5194/amt-13-5193-2020). Many details on radar simulation are extensively described by literature.

While these are useful tools, CR-Sim and PAMTRA both lack the ability to perform multiple scattering calculations for the radar, which is essential to capturing realistic radar propagation from a spaceborne platform at such high millimeter-wave frequencies. Furthermore, we don't believe that the existence of other radar simulators in the literature precludes the development of a custom simulator for a specific/targeted application or study. Indeed, it is still common practice to describe the ingredients of a millimeter-wave radar forward simulator without providing an exhaustive comparison to other such simulators already described in the literature. This can be found, for instance, in a recent paper on spaceborne radar simulation published in the same journal (Battaglia, A., Kollias, P., Dhillon, R., Lamer, K., Khairoutdinov, M., and Watters, D.: Mind the gap – Part 2: Improving quantitative estimates of cloud and rain water path in oceanic warm rain using spaceborne radars, Atmos. Meas. Tech., 13, 4865–4883, https://doi.org/10.5194/amt-13-4865-2020, 2020.)

Thus, the authors should

(i) move technical details in Sec. 2.2. to the appendix;

We have moved the contents of section 2.2.1, which details the single-scattering calculations and microphysical parameterizations, to Appendix A. However, we believe sections 2.2.2 and 2.2.3 belong in the main text, and so have not moved these sections, but removed the subsection titles so that all of this content now falls under a single Section 2.2. The contents of section 2.2.2, which is only a single paragraph, detail the interpolation procedure used to oversample the radar range axis, which is important context for understanding the simulation results. The contents of section 2.2.3 include specification of the high-level radar parameters, spatial averaging scheme, and the uncertainty model, and again are important for understanding and interpreting the results presented in Section 3.

(ii) reduce their selection of Figures 5-12; e.g. by compressing the presentation of the simulated measurements and/or moving them to Sec. 2.2.3 or the appendix

These figures constitute the main results of the paper, which are the analyses of end-to-end DAR simulations performed on 5 distinct LES cases. We believe that the amount of detail presented is warranted given the scope of the study, and therefore do not believe that these figures should be compressed or moved within the manuscript.

***Specific comments:***
1. Two relevant contributions to this field are missing in the introduction: Lamer et al (2021), doi.org/10.5194/amt-14-3615-2021 ; Schnitt et al (2020) , doi.org/10.1175/JTECH-D-19-0110.1
We thank the reviewer for pointing these out. We have included references to these two publications in the introduction.

2. p.3, ll. 16 More details should be provided on selection of 155.5GHz channel as third frequency. Why was the channel at 167 GHz (Roy et al, 2020) moved to 168 GHz here?
Please see response to General Comment #4 above for the reasoning behind the 155.5 GHz channel. Given a minimum (155.5 GHz) and maximum (174.8 GHz) DAR channel location, there is freedom in choosing where the 3rd DAR frequency should go in the interval (155.5, 174.8). We chose the value of 168 GHz because it is the DAR frequency that minimizes the correlation in the PBL between the fit parameters $a_2$ and $a_3$ in Eq. (6) and (7) of the revised manuscript. We note that the difference in this correlation between 167 and 168 GHz is not large.

3. p.4, tab. 1: the following LES parameters should be added: vertical top height of LES simulation; longitude and latitude of domain center; considered time step; IWV variability within the domain. I suggest to indicate mean and standard deviation. IWV (or TCWV?) should be used consistently throughout the manuscript. I also suggest to quantify the variability of $\rho v$ and IWV in the respective simulations to give the reader a feeling on the expected variations of the respective moisture fields.

The requested values for each LES case have been added to Table 1.

4. p.4, ll.5: A clarifying sentence should be added on how the different horizontal and vertical resolutions of LES and MERRA-2 were accounted for when completing the LES columns. A reference and specifications to the used MERRA-2 dataset are missing.

We note that in the original manuscript, it is stated that we utilize one-dimensional MERRA-2 reanalysis data for completing the LES columns, and so the horizontal resolution difference is not necessarily relevant. In the revised manuscript, we have provided a proper reference for the MERRA-2 data set, and have added a sentence listing the horizontal and vertical resolutions for MERRA-2.

5.p.4, Sec. 2.2: also see major comment 5. The authors should highlight the advantages of their radar simulator compared to existing radar forward models.

We point the reviewer to our comments under major comment 5.

6.  p.8, l.17: Which 'relevant variables' were chosen?

We thank the reviewer for pointing out this confusing phrase. All that is meant here is that the integrand of old Eq. 4 (now Eq. 1 in the revised manuscript) is summed at the discrete model points. We have removed the phrase "relevant variables" in the revised manuscript to make it clear that we're simply performing the discrete version of the integral defined in that equation.

7.  p.9, tab.3: range indication for *dBZmin* and explanation for * in horizontal footprint line are missing.

We have clarified in the table caption that the minimum detectable reflectivity corresponds to the surface range. The asterisk in the horizontal footprint line was a typo in the original manuscript and has been removed in the revised version.

8.  p.11, Fig.2: I suggest to add the reflectivity profiles of f0 and f2 to the right panels of (a) and (b), respectively. Considering a detection threshold of -34dBZ as given in Tab 3, a detection of the profile as shown in (b) below 2km does not seem possible. A clarifying statement should be added on how saturated profiles are dealt with in the retrieval, and how often saturation occurs in the 5 different LES cases. How many profiles were simulated in total? CFAD diagrams could give the reader an additional overview on the simulated radar measurements.

We first note that the point of this figure is to investigate in a "toy model" fashion the bias differences between the 2-frequency and 3-frequency DAR approach, and we specifically discuss in the text (Pg. 11 line 5-8 of the original manuscript) that this analysis is not performed with the full radar simulator, but in a simplified model to only highlight that these biases exist even for a "perfect" measurement. To clarify this point further, we have edited the second sentence of the caption to Fig. 2 as follows (new text in italics): "An ideal reflectivity measurement is assumed, with no multiple scattering, arbitrarily high range resolution *and measurement sensitivity*, and zero random measurement error."

The detection threshold is very important for the main simulation results in the paper, and is applied appropriately therein as can be easily seen from the reflectivity profiles in Fig. 3(d).

We're not sure what the reviewer means by "saturated profiles", but are assuming that they are getting at situations where the beam is heavily attenuated by liquid and water vapor. In the simulations, we account for this reality by removing any reflectivity observations below the detection floor, as explained

Finally, we believe that a CFAD-style analysis is well outside the scope of this toy-model-like investigation that is not part of the results section of the paper.

9. p.15, ll.11: the authors should clarify how exactly they chose the profiles that were sampled when applying intelligent scanning sampling strategy.

We thank the reviewer for pointing out that we did not explicitly state this criteria. We have added text to clarify this point.

Added Text page 14 line 3: "In this work, we assume that for each along-track pixel the antenna scans to the cross-track location with maximum liquid water path."

10. p.15, ll.11: Do the authors think that an intelligent scanning technique could be, in a realistic application, combined with adjusting one of the frequencies to overcome sensor saturation for the expected conditions?

It is an interesting comment and is something that we have thought about. There is no technical reason why this could not be done – however it would be somewhat challenging to know given some ancillary measurement (we envision a forward-looking radiometer) to know exactly which frequency, or pulse shape, etc. would be best suited for each target. Our guess is that the complexity (=cost) involved in going down this path would not be justified by the return on that investment. At this time if we can just avoid the clear-sky we would be happy. We leave this discussion out of the manuscript as it is not something that we are likely to pursue and is a bit distracting from the story.

11. p.16, Fig.3: figure caption for panel (d) is missing.

We thank the reviewer for catching that this caption was missing. We have included the panel (d) description in the figure caption in the revised manuscript.

12. p.17, ll.2; Fig. 4(c) (and 6,8,10d): The authors should add a clarifying comment how the $\rho v$ precision changes at different heights (e.g. lower BL below 3.9km in the GATE case), or why the specific shown height was chosen. I suggest to add a clarifying statement, or to add the precision at different heights to the respective panel.

The heights used for this precision scaling analysis are chosen such that they lie well within the average cloud/precip vertical extent, and therefore have a good amount of radar sampling. Of course, one could always choose a more pessimistic case near the cloud edge, and the degradation of this retrieval precision as a function of vertical position can be clearly seen in the error bars of the (a) panels for the respective cases/figures.

Added Text to Fig 3 (Fig 4 from the old manuscript) caption: "For each LES case, this profiling height is chosen to lie well within the average cloud and precipitation volume to avoid sparse sampling near the lower and upper boundaries."

13. p.24, Fig.13: This figure summarizes the results of the intelligently scanning method nicely. How do these results differ when applying the analysis to the nadir sampling strategy? I suggest to address this in the text to uphold figure clarity. The authors should add to their discussion whether the derived along track averaging distance is sufficient to resolve the expected horizontal water vapor variability.

As one would expect, the retrieval performance generally degrades when going from the intelligently scanned to fixed-nadir system, though it does depend on the specific case (e.g., in regimes with high horizontal homogeneity it has a small impact). The main impact in going to the fixed-nadir system is that the error bars increase in a manner that can be clearly identified from the panel (d) plots in figures 4, 6, 8, 10, and 12. Importantly, the biases are quite similar, are generally small, and can be identified readily from the panel (a) plots in those same retrieval analysis figures.

For clarity we have added the following text to the figure caption: "While the retrieval biases are nearly identical for the fixed-nadir case with large along-track averaging, the precision is degraded and can be assessed by examining the precision scaling plots in Fig. 3(d), 5(d), 7(d), 9(d), and 11(d)."

14. p.25, ll.18: I am wondering if the required fulfillment of the 200m vertical resolution systematically excludes certain conditions – _can the authors provide a threshold or typical cases in which this requirement is not fulfilled?

As the retrieval is formulated in the current work, the requirement of 200 m vertical resolution in the retrieval is by definition always fulfilled for the "in-cloud" retrievals. Unlike for passive systems, the vertical resolution of the radar measurement and retrieval are perfectly well understood thanks to the range-gating nature of radar. Of course, it will certainly be the case in some regimes and at some vertical levels that there is not enough differential absorption signal or measurement sensitivity to perform the 200 m retrieval at the required precision, and the results of this study give a quantitative understanding of this for different cloud regimes.

15. p.26, tab. 4: A column giving the surface temperature should be added.

This information has been added to Table 1.

16. p.26 ll.20: A reference for this statement should be added.

We have appended the following text in the revised manuscript and added a relevant reference for this statement: "due to the large Rossby radius of deformation".

Reference: Schneider, E. K., 1977: Axially symmetric steady-state models of the basic state for instability and climate studies. Part II. Nonlinear calculations. *J. Atmos. Sci.,***34,** 280–296.

16. p.27, l.15: the authors should provide more information about the applied LWC > 0.01gm-3 threshold and the in-cloud temperature derivation for thinner clouds. How does this threshold impact the presented results?

The LWC threshold only applies to analysis of the model variables themselves, and not the DAR-inferred temperature profiles (i.e., varying this threshold will not change the DAR results at all since it is not applied to them). While the exact values for the mean model RH and T profiles will vary slightly with this threshold, the take-home point of the figure is clear - that estimating temperature based on knowledge of the water vapor density and an assumption of RH = 100% will often lead to an underestimate of the temperature, or low bias. Reasonable adjustments of this threshold will not alter this conclusion.

17. p.27: Fig 14b) is referenced before 14a).

In the revised manuscript we have edited the initial reference to this figure to simply state "Fig. 14" instead of "Fig. 14(a)", since this reference introduces the figure as a whole. Later we reference 14(a) to explain that these RH profiles are used as input to the derived temperature profiles in 14(b), which is the natural flow for the figure.

18. p.28, conclusions: based on their experience with the present analysis, I encourage the authors to add a concluding remark how their simulated system would perform in different conditions such as at higher latitudes.

We have added the following text to the end of the conclusions section: 'While this study chose the subtropical to tropical marine cloud transition to evaluate the potential capabilities of a spaceborne DAR we can speculate as to the expected performance in different cloud regimes. We expect two primary limitations. First, we expect significant degradation of the radar sampling due to attenuation in the most heavily precipitating storms (e.g. mesoscale convective systems, Hurricanes, and strong mid-latitude frontal systems. Second, relative to the subtropical ocean many land regions will have insufficient scattered cumulus to acquire significant samples to perform vapor retrievals even with a scanning system. On the other hand, we expect that DAR retrievals will be quite useful in the midlatitude storm track regions over ocean in regions of post frontal convection and stratocumulus which are frequently multi-layered and lightly precipitating thereby providing ideal targets. In addition, cloudy polar and high latitude scenarios will be particularly well suited to the DAR technique because of the relatively small amount of attenuation that the radar would experience.'

---

## Author Comment (AC2)

Review of manuscript #AMT-2021-111 entitled 'Spaceborne differential absorption radar water vapor retrieval capabilities in tropical and subtropical boundary layer cloud regimes' by Richard Roy, Matthew Lebsock, and Marcin Kurowski.

The manuscript provides a detailed evaluation of the potential of a differential absorption radar (DAR) operating in three tones around the 183 GHz water vapor absorption line for PBL humidity profiling inside clouds, precipitation and for measuring integrated water vapor (IWV) in clear air regions. While the concepts presented by the authors are innovative, I believe the study overlooks a few effects that would have a considerable impact on the results (see major comments below). Given this, my recommendation is to reject the manuscript in its current form and invite the authors to submit a revised manuscript when these effects are simulated.

**Major comments:**

1.  I believe the authors have omitted to consider the spaceborne range weighting function in their forward simulations. Past literature has shown that the range weighting function of long pulsed radars effectively stretches along, or in other words vertically smooths, any vertically-narrow cloud or precipitation feature. I believe this would affect the forward simulated radar observables that the authors present and rely on to assess their technique. Beyond vertically stretching/smoothing the fields, the range weighting function should be expected to introduce covariance of the random error. The authors correctly introduce random error after the retrievals are performed based on the linearity of the transfer function from the state vector to observations (Page 15); however, I believe this only represents one source of the true error, which should also include error caused by the covariances along range especially if the observations are to be oversampled by a factor of 4 of the true radar resolution to achieve the desired range gate spacing.

    The reviewer is correct that effects on the water vapor retrievals coming from a specific range weighting function (RWF) are not included in this work. While we will address the individual points mentioned above, we first note that our approach is equivalent to assuming a RWF that is a top-hat function of width 50 m. This is useful from a measurement capability assessment point of view because it does not assume specific implementation-level aspects of the transmit waveform and pulse compression filter. Another way of specifying the RWF properties that would be applicable to this study is to assert that, regardless of the chosen waveform parameters, the RWF must ensure negligible random error covariance between range bins that are separated by 50 m. While this is admittedly idealized in some respects, we still find it useful for the purposes of this study as we're not interested in analyzing a particular implementation-specific architecture, but rather examining the fundamental limits of the DAR technique for PBL humidity observations.

    Another point to clarify before addressing the main point of the reviewer's main comment here is the effect of range oversampling of the retrieval. The resolution that is oversampled here is the water vapor profile resolution of 200 m, which is not inherently related to the range resolution or RWF. In terms of the radar signal processing, we are assuming that the radar echo power vs. range profile is sampled at exactly the range

resolution of 50 m (i.e., neither undersampled nor oversampled). Therefore, it should NOT be thought of as oversampling the RWF by a factor of 4, which would lead to significant bin-to-bin covariance terms, but rather adequately sampling the RWF while purposefully downsampling when performing the water vapor retrieval.

To begin addressing the reviewer's main point, we agree that the effect of an RWF on long-duration-pulse radar observations is to stretch/smooth features in the vertical, where the relevant length scale for the "smoothing" is the width of the RWF itself. The degree to which this smoothing will become problematic for the retrieval requires comparing the characteristic (vertical) length scale over which the cloud/precipitation reflectivity profiles change appreciably with the width of the RWF. This is of course a detailed task that depends sensitively on both the cloud/precipitation regime and the implementation-specific RWF, and so we don't believe that it belongs in this manuscript which is focused on addressing higher-level questions of the spaceborne DAR capabilities. However, we do provide in this response a detailed assessment of the RWF impacts on retrieval biases for a specific profile from the RICO case to highlight that, with sufficient pulse chirp bandwidth, the impact of bin-to-bin correlations on the retrievals is minimal. This discussion is provided here for the reviewers, however we do not believe that this material is appropriate for the manuscript.

We begin our RICO RWF case study by specifying a few candidate transmit waveforms, and by using the matched filter as the pulse compression filter. To match the notional high-level parameters used in the main manuscript, we assume a pulse length of 50 us, a Hann window for the transmit waveform taper, and a list of candidate chirp bandwidths: $B = 0.75, 1.5, 3, 6,$ and 12 MHz. We then calculate the RWF, as well as two parameters related to the width of the RWF: (1) the Fourier limited resolution in range, $c/(2B)$; and (2) the effective range resolution calculated as the equivalent width of a top-hat RWF with the same area as the true RWF. This latter range resolution (2) is more representative of the true range resolution of the system, and is exactly the range resolution term that arises in the weather radar (or volumetric scattering radar) equation. The plot below describes the difference between these two parameters, plotted against the compressed pulse time axis, which can be converted to range by multiplying by the factor $c/2$.

[Figure]

Below we plot the individual RWFs for each candidate chirp bandwidth, as well as provide a table of the relevant range resolution parameters.

[Figure]

| B (MHz) | c/2B | cT$_{res}$/2 |
|---------|---------|---------|
| 0.75 | 200 m | 389 m |
| 1.5 | 100 m | 194 m |
| 3 | 50 m | 97 m |
| 6 | 25 m | 49 m |
| 12 | 12.5 m | 24 m |

The next task is to create simulated reflectivity profiles for each RWF (and each of the 3 DAR channels) and perform a retrieval in each case. Here we do so by working in the single-scattering approximation, and by convolving the simulated, attenuated reflectivity profile (which is at the very high LES vertical resolution) with the RWF, and downsampling the resulting convolved profile at the cT$_{res}$/2 effective range resolution (Shown in Figure below). Then, we perform the water vapor retrieval using the treatment outlined in the "DAR Measurement Methodology" Section of the manuscript (new Section 2.3). Examples of the convolved reflectivity profiles before downsampling are shown below for the 155.5 GHz DAR channel:

[Figure]

Note that we have chosen a profile from RICO that contains a sharp cloud/precipitation feature just above 2 km, which is the type of feature that could result in RWF-induced biases in the water vapor retrieval for coarse enough range resolution (i.e., small chirp bandwidth). Furthermore, example downsampled (solid circles) profiles at the 3 DAR frequencies for 2 different chirp bandwidths are shown below:

[Figure]

Note that we do not add random error/noise to these profiles, though as the reviewer highlighted it can in theory be introduced after the retrievals are performed based on the linearity of the transfer function from the state vector to observations. Instead, what is most important to assess from a bin-to-bin covariance point of view is the impact of the RWF convolution on retrieval biases (i.e., systematic error). The resulting retrievals for each chirp bandwidth are shown below, where we've used a retrieval step size of $r^+-r^- = 200$ m (see Section 2.3):

[Figure]

As expected, the low chirp bandwidths of 0.75 and 1.5 MHz, which either undersample (0.75) or provide commensurate sampling (1.5) of the retrieval resolution of 200 m, result in significant biases, especially in the vicinity of the strong reflectivity feature above 2 km. However, for larger chirp bandwidths, the stretching effects of the RWF convolution do not affect the retrieval accuracy. Importantly, the range resolution assumed in our simulation study in the manuscript of 50 m corresponds to a chirp bandwidth of 6 MHz, which is well into the regime where the RWF stretching does not affect the retrieval.

This case study shows that, even for a pessimistic scenario where there are sharp features in the reflectivity profile, the assumed radar range resolution of 50 m is consistent with a 6 MHz chirp and resulting RWF that limits bin-to-bin covariance to a level where it does

not significantly impact the retrieved humidity accuracy. While this does not prove the independence of our analysis and results on the specifics of the RWF for all possible measurement scenarios and cloud/precip regimes, as there could be contrived examples where even the 6 MHz RWF results in bias issues, we believe it adequately addresses the reviewer's concern around the stretching from the RWF.

We do note here that, in the original manuscript on pg. 3 line 22, we state that a range resolution of 50 m is consistent with a pulse bandwidth of 3 MHz, which was incorrect. This is only the case for the $c/2B$ figure, and for the purposes of the radar volumetric scattering sensitivity, it is more accurate to use the $cT_{res}/2$ figure. Therefore, in the revised manuscript we have edited this to read 6 MHz. Additionally, we have added a sentence immediately after this line acknowledging that these results do not include specific aspects of the RWF, and that our approach is equivalent to assuming a top-hat RWF with a width 50 m. Finally, we have added a sentence in the conclusions section (end of the first paragraph) noting that future investigation is needed to assess the impact of these biases for specific instrument implementations.

Added Text Page 3 Line 28: "We note that the approach implemented here does not include specific aspects of the radar range weighting function (RWF), and is instead equivalent to assuming an RWF that is a top-hat function of width 50 m."

Added Text Page 28 Line 1: "Future investigations focused on specific instrument architectures and radar waveform parameters are necessary to fully assess the impact of range-bin-to-range-bin correlations introduced by the range weighting function."

2. The authors discuss the use of a spaceborne DAR system operating a long pulse with frequency modulation like that used in RainCube. Based on my limited knowledge of RainCube's performance, I would expect that such a radar would suffer from contamination by the surface echo and range side lobes in the lowest kilometer of observations above the surface. Yet the authors failed to mention anything about how the proposed radar would be impacted by these two effects (surface echo and range side lobes) and about how this may impact the sensitivity of this radar to low level targets and the overall performance of the proposed retrieval in the boundary layer.

The reviewer is correct that radars using pulse compression with large time-bandwidth products will generally have to contend with all of the impacts associated with the range weighting function. In the preceding major comment, we addressed this in the context of range-bin-to-range-bin covariance and potential impacts on the water vapor retrieval. As the reviewer points out, another potential issue stemming from the RWF is the surface response and its impact on the radar's sensitivity, especially in the lowest layers of the atmosphere. The surface response involves the combination of the beam pattern (with its own sidelobes), the RWF, and the projection of the beam onto a spherical surface, which results in surface returns that are spread over range. For a system like RainCube, this generally limits the detection of precipitation to heights above 500 m (see discussion in Battaglia et al., Spaceborne Cloud and Precipitation Radars: Status, Challenges, and Ways Forward, *Reviews of Geophysics*, 2020), though this neglects the added sidelobe suppression from beam attenuation in liquid. However, the surface clutter for the notional

radar described here is greatly reduced relative to RainCube for two main reasons: (1) the chirp bandwidth in our case is (6 MHz)/(2.5 MHz) = 2.4 times larger; and (2) because of the shorter pulse duration relative to the pulse coherence time. In our case, we use a pulse duration that is around half the "time to independence", whereas in RainCube the pulse time of 166 us is significantly longer than the time to independence of around 25 us.

The preceding discussion was intended to provide context for why the situation here is much different than the only reference case for a long duration pulse radar (Raincube). To quantitatively assess the difference, it is necessary to implement a surface scattering model that includes a realistic beam pattern, surface illumination geometry, Doppler range migration, and the RWF. While we do not believe this level of detail belongs in the manuscript, we provide such a calculation here in response to this particular point. The surface clutter profile below assumes a realistic beam pattern resulting from the tapered illumination of a parabolic reflector with a Gaussian feed horn. For these simulations, we use the taper value of 10.9 dB that maximizes aperture efficiency. Furthermore, we use the same pulse parameters described above: 50 us pulse with 6 MHz chirp bandwidth and a Hann window. Guided by recent aircraft studies with JPLs Vapor In-cloud Profiling Radar (VIPR) instrument, we assume a surface normalized radar cross section (NRCS) of 15 dB, which is a pessimistically large value corresponding to very low wind speed conditions at near-nadir incidence. We perform the calculation at the middle DAR channel frequency of 168 GHz. Furthermore, we note that if we use the high-level parameters from the RainCube instrument in our surface clutter simulation (0.5 m aperture, 166 us pulse, 2.5 MHz chirp bandwidth), we reproduce the surface clutter suppression (~65 dB at 0.5 km above the surface) as modeled in Beauchamp et al., Observations and Design Considerations for Spaceborne Pulse Compression Weather Radar, *IEEE TGRS*, 2020 (https://ieeexplore.ieee.org/document/9169707).

[Figure]

In the plot above, negative range corresponds to the atmospheric region of interest. We find that the clutter profile intersects the minimum detectable reflectivity (see Table 2 of the revised manuscript) of -34 dBZ just below 200 m above the surface. Therefore, we

once again see that the large chirp bandwidth of 6 MHz (and corresponding fine range resolution of 50 m) is necessary to mitigate a variety of factors, including surface clutter, that could potentially limit DAR water vapor retrieval sensitivity and coverage. In short, surface clutter is not a limiting factor for the studies presented here because of the very high range resolution used.

We have added a short discussion of surface clutter at the end of the "Radar Forward Model" Section.

Added Text Page 8 Line 5: "In addition to thermal noise, surface clutter can potentially limit the radar detection sensitivity in the lowest layers of the atmosphere. In this case, we find that the large pulse bandwidth of 6 MHz and pulse duration that is well below the decoherence time result in a surface clutter profile that is largely confined to the lowest few hundred meters above the surface, with the surface equivalent reflectivity falling below -40 dBZ at a height of 200 m. Therefore, we do not expect this clutter source to limit detection sensitivity. We note that this contrasts with the experience of RainCube, where measurement sensitivity is degraded in the lowest 0.5-1 km of the atmosphere because of surface clutter (Battaglia et al., 2020). However, key differences between the two systems that explain this difference are the relative chirp bandwidths and the ratio of pulse duration to coherence time, which in the case of RainCube is larger than 1."

3. While it is interesting to consider the use of intelligent sampling using a passive/active sensor synergy, I believe the authors need to discuss more of the key aspects of such an endeavor to prove that it would even be feasible. For one, the authors did not discuss how a 2-m dish antenna will be configured to perform intelligent pointing. Overall, I believe a lot more information needs to be presented regarding this theoretical intelligent scanning technique. I would even argue that it should be the topic of a separate study.

Intelligent scanning of cloud radars towards high-value targets is not a new idea, and there are several methods that could be technologically feasible for accomplishing this. The most obvious antenna solution that has airborne heritage (e.g., AirMASTR) is a phased-array fed reflector (PAFR), which in this case could consist of a line feed illuminating a 1D parabolic reflector. Such a system would easily be able to point in an agile fashion across the field of regard discussed in this work. Another potential solution is a mechanically steerable sub-reflector that illuminates a primary slightly off the paraboloid axis. This technique is routinely used in quasioptical systems with parabolic reflectors to achieve steerability. However, we believe that these details are beyond the scope of the current work, which is primarily concerned with answering the following question: If such idealized radar performance can be assumed, what is the quality of the water vapor retrievals that result from such a system in these various scenarios? We agree with the reviewer that the details of this implementation ought to be the topic of a separate study, and there are many implementation-specific problems to be solved. However, this paper is specifically not focused on the implementation-specific problems, and more on the basic physical limitations of DAR water vapor retrievals from space.

**Minor comments:**

1. P2, L14: The authors claim that they performed detailed orbital simulations. Please elaborate on what detailed information about the satellite orbit is used in this study beside the satellite altitude and velocity.

   For clarity we have changed this to 'detailed radar simulations from an orbital altitude.'

2. P3, L31: I think the word "Table" is missing in front of "1".

   The word "Table" was indeed missing. We have inserted this in the revised manuscript.

3. P6, L1: A formal study has been published based on Tatarevic et al., 2019 report. As such, I believe it is more appropriate to cite Oue, M., Tatarevic, A., Kollias, P., Wang, D., Yu, K., and Vogelmann, A. M.: The Cloud-resolving model Radar SIMulator (CR-SIM) Version 3.3: description and applications of a virtual observatory, Geosci. Model Dev., 13, 1975–1998, https://doi.org/10.5194/gmd-13-1975-2020, 2020

   Thank you for pointing us to this reference. We have replaced the reference to the report with the formal study reference.

4. P7, Fig. 1: Why did the authors not consider the Self-Similar Rayleigh-Gans Approximation (SSRGA) for the snow radar forward scattering modeling. Here is a useful reference: Hogan and Westbrook, 2014: Equation for the Microwave Backscatter Cross Section of Aggregate Snowflakes Using the Self-Similar Rayleigh-Gans Approximation, Journal of the Atmospheric Sciences, 71(9), 3292-3301.

   We are aware of the use of the self-similar Rayleigh-Gans approximation for modeling mm-wave scattering from snow aggregates, and there is no fundamental reason why this could not have been used in our study. However, SSRGA only applies to large aggregates consisting of many individual monomers since it relies on a statistical/ensemble-based decomposition of the particle mass distribution in Fourier space. Therefore, it would not apply to the pristine "ice" species in this work. We note that the main purpose for treating ice and snow scattering in this work is to assess the degree to which these particles introduce a bias in the DAR retrieval due to frequency-dependent scattering (i.e., that which is associated with the parameter $a_2$ in Eq. (10)). Specifically, note that we are not attempting to model the G-band radar's ability to retrieve ice microphysical parameters, we simply want an approach that provides a realistic quantification of the frequency-dependence of the scattering. Since SSRGA is derived assuming an ensemble mean approach, there is good reason to believe that non-trivial frequency dependence that might be present in the single crystal realization case may be "washed out" by performing the ensemble average, but we have not done a detailed study of this (well beyond the current scope). It could be interesting in future work to investigate the different DAR biases predicted using the SSRGA and DDA formalisms, but this is certainly beyond the scope of the current work, where we're primarily focused on PBL water vapor and therefore liquid hydrometeors.

5. P9, Fig. 2: Is it possible that the precipitation and cloud mixing ratio labels are mixed up in the right panel of figure a? I find it difficult to believe that the rain is elevated and higher in magnitude than the cloud mixing ratio.

The labels in the figure are correct. A potential explanation is the presence of shear near cloud edges and the fact that rain has its own terminal velocity, which can separate it from the cloud regions. For such conditions, turbulent eddies can transport rain water away from the cloud at larger distances because its lifetime is longer than for cloud water in unsaturated regions. Cloud water evaporates quickly in unsaturated environments due to the bulk saturation approach, which is commonly used in LES modeling as an approximation of phase changes for smaller droplets. Rain water can exist longer because the evaporation rate is slower.

6. P9, Fig. 2: I would recommend not showing the water vapor retrievals before presenting section 2 that describes the retrieval.

The water vapor retrievals shown in this figure do not use the new retrieval algorithm detailed in Section 2.4, but rather the simplified humidity retrieval algorithms discussed in 2.3 (Eqs. 9 and 10). That is, this figure is positioned within the same section that describes the algorithms used, and therefore we deem it appropriate. We have added a reference to the appropriate section 2.3 in the figure caption to clarify this.

7. P16, Fig. 3: The caption for panel d is missing.

Thank you for catching this missing caption. We have corrected this in the revised manuscript.

8. P18, Fig. 5: I believe the surface echo and range side lobe should be visible in this figure. Afterall this is a chirp radar and its sensitivity is very high (better than CloudSat) and as such I would expect contamination of at least all the sub-cloud layer.

Please see response to Major Comment 2 above. In short, we treat the surface response with associated range sidelobes as a height-dependent noise floor, which is subtracted off of the detected power vs. range profiles to arrive at a simulated observation of reflectivity (which should have no contribution from the thermal noise floor or from range sidelobes). Because of the high chirp bandwidth of 6 MHz, these range sidelobes are limited to the first 200 meters above the surface. This is one of the main reasons for choosing such a high range resolution - i.e., to limit the impact of range sidelobes on PBL observations.

9. Can you comment on the ability of the technique to capture the mesoscale organization of the humidity field?

The main limitation in a DAR's ability to capture this sort of organization in the humidity field is the requisite horizontal averaging necessary to reduce measurement uncertainty to a sub-variability level. Practically, we don't see this as being possible for the profiling measurement. If one is focused on the total column water vapor retrieval, which can achieve low uncertainty with much less horizontal averaging, such a study could be

feasible. However, it likely requires a full cross-track scanning system to resolve the 2D humidity field, which is not the concept considered here.

10. P21, F9: I believe the stretching effect of the range weighting function should be visible in this figure. A useful reference to visualize this effect is Lamer, K., Kollias, P., Battaglia, A., & Preval, S. (2020). Mind the gap–Part 1: Accurately locating warm marine boundary layer clouds and precipitation using spaceborne radars. AMT, 3(5), 2363-2379.

    Please see response to Major Comment 1. In short, our approach is equivalent to assuming a range weighting function (RWF) that is a top-hat function with a width given by the range resolution of 50 m. All spaceborne radars deployed to-date have utilized pulse parameters that result in much broader range weighting functions (and therefore range resolution), and so for those systems you would expect to see significant "stretching" or "blurring" of localized cloud features like these.

    In terms of justifying the top-hat RWF assumption, for the scope of questions we are interested in addressing in this work, we believe this is the best approach since it does not require a specific choice of radar waveform parameters (e.g., waveform taper/window), though we do provide some notional ideas of things like transmit power and chirp bandwidth. In reality, a mission would go through a significant process of analyzing and optimizing the choice of all Tx and Rx parameters, and we are not interested in assessing a single detailed architecture, but rather getting higher-level answers to basic questions about the retrievability of PBL water vapor from space using the DAR technique.